# A Scoping Review of AI-Based Approaches for Detecting Autism Traits Using Voice and Behavioral Data

**DOI:** 10.3390/bioengineering12111136

**Published:** 2025-10-22

**Authors:** Hajarimino Rakotomanana, Ghazal Rouhafzay

**Affiliations:** Department of Computer Science, Université de Moncton, Moncton, NB E1A 3E9, Canada; ehr9507@umoncton.ca

**Keywords:** Autism Spectrum Disorder (ASD), machine learning, voice biomarkers, movement analysis, motor skills, facial gesture, eye gaze, visual attention, activity recognition, speech features, multimodal AI

## Abstract

This scoping review systematically maps the rapidly evolving application of Artificial Intelligence (AI) in Autism Spectrum Disorder (ASD) diagnostics, specifically focusing on computational behavioral phenotyping. Recognizing that observable traits like speech and movement are critical for early, timely intervention, the study synthesizes AI’s use across eight key behavioral modalities. These include voice biomarkers, conversational dynamics, linguistic analysis, movement analysis, activity recognition, facial gestures, visual attention, and multimodal approaches. The review analyzed 158 studies published between 2015 and 2025, revealing that modern Machine Learning and Deep Learning techniques demonstrate highly promising diagnostic performance in controlled environments, with reported accuracies of up to 99%. Despite this significant capability, the review identifies critical challenges that impede clinical implementation and generalizability. These persistent limitations include pervasive issues with dataset heterogeneity, gender bias in samples, and small overall sample sizes. By detailing the current landscape of observable data types, computational methodologies, and available datasets, this work establishes a comprehensive overview of AI’s current strengths and fundamental weaknesses in ASD diagnosis. The article concludes by providing actionable recommendations aimed at guiding future research toward developing diagnostic solutions that are more inclusive, generalizable, and ultimately applicable in clinical settings.

## 1. Introduction

### 1.1. Background

Autism Spectrum Disorder (ASD) is a complex neurodevelopmental condition that affects communication, behavior, and social interactions. The early diagnosis of autism is crucial for improving long-term outcomes, as it enables timely therapeutic interventions. The Diagnostic and Statistical Manual of Mental Disorders, Fifth Edition–Text Revision (DSM-5-TR), released in March 2022, remains the official guideline for the evaluation and diagnosis of ASD. However, the process can be time-consuming, resource-intensive, and highly dependent on clinical expertise. According to the Public Health Agency of Canada, published in 2022 [1], 1 in 50 (2%) of Canadian children and youth aged 1 to 17 years were diagnosed with ASD. The findings also reveal that only 53.7% of these cases were clinically identified before the age of five. In fact, based on the DSM-5 criteria, using a diagnostic tool, the Autism Diagnostic Observation Schedule (ADOS), individuals on the autism spectrum are evaluated based on the following two main aspects of development:Social communication and social interaction deficits.Restrictive or repetitive behaviors (RRBs), interests, or activities.

Recent advances in artificial intelligence (AI) have facilitated the development of innovative tools for the early detection of autism traits that analyze various data modalities, including voice characteristics [2], MRI/fMRI and brain imaging, electroencephalography (EEG) signals, eye-tracking [3], facial expressions [4], and both verbal and non-verbal cues [5]. Similar AI-based approaches have also been explored to support the understanding and detection of other neurodevelopmental conditions through behavioral datasets, such as Attention-Deficit Hyperactivity Disorder (ADHD) [6] and Cerebral Palsy [7], as well as to assist in the assessment of psychiatric disorders [8], including schizophrenia, anxiety disorders, depression, and bipolar disorder.

### 1.2. Objective of the Review

This scoping review explores the growing body of research on AI-driven techniques for detecting autism traits through voice and behavioral data. More precisely, AI-based approaches focus predominantly on the following eight key modalities: “voice biomarkers”, “ conversational/interactional dynamics”, “linguistic and content-level language analysis”, “movement analysis”, “Activity recognition”, “facial gesture analysis”, “visual attention”, and “multimodal approaches”. By mapping the current landscape, we aim to highlight the potential of these innovative approaches in supporting the earlier, more accessible, and objective detection of ASD.

The following are the main research questions that were defined to guide the review:What is the performance of the models applied in the research to detect autism traits and what modalities have been explored?How to extract voice and behavioral data features to integrate into existing models?What are the challenges and limitations of the approach using voice and behavioral data?What are the ethical constraints for research on people with ASD?

### 1.3. ASD Discrimination Criteria from a Clinical Point of View

The eight modalities reviewed were selected to support early, accessible, and effective screening tools that can enhance clinical practice. They focus on behavioural features of ASD that can be assessed without medical imaging or genetic testing, similar to those evaluated through clinician observations in tools such as the Autism Diagnostic Observation Schedule (ADOS). Before exploring the studies on AI-driven analysis of voice and behavioural data, it’s important to develop a clear understanding of the unique features each modality offers for identifying ASD. This section presents the categorized modalities and explains how they differ from one another.

#### 1.3.1. Voice Biomarkers

This modality quantifies the prosodic and acoustic anomalies often described clinically as “odd vocal quality.” The key discriminative features include the following:Atypical Prosody:Measured as reduced pitch (fundamental frequency F0) variation, leading to a monotonic or flat-sounding voice that lacks the melodic contours used for emotional emphasis and questioning.Vocal Instability: Increased jitter (frequency instability) and shimmer (amplitude instability) can manifest as a shaky, rough, or strained voice quality, reflecting poor neuromotor control of the larynx.Abnormal Resonance and Timbre: Deviations in formant frequencies and spectral energy distribution can make the voice sound atypically nasal, hoarse, or strained.

#### 1.3.2. Conversational/Interactional Dynamics

This modality provides a direct measure of “deficits in developing, maintaining, and understanding relationships” and “difficulties adjusting behavior to suit various social contexts” by analyzing the micro-rules of social dialogue. The key discriminative features include the following:Atypical Turn-Taking: Characterized by either excessively long response latencies, suggesting difficulty with rapid social processing, or frequent turn-taking overlaps and interruptions, indicating challenges with social timing and reciprocity.Lack of Prosodic Entrainment: Failure to subconsciously synchronize pitch, intensity, or speaking rate with a conversation partner. This absence of “vocal mirroring” is quantifiable evidence of deficits in social-emotional reciprocity.Dysfluent Pause Structure: Irregular and unpredictable pausing patterns disrupt the natural flow of conversation.

#### 1.3.3. Linguistic and Content-Level Language Analysis

This modality focuses on what is said, identifying patterns in language structure and content. Clinical discrimination criteria include the following:Pragmatic Language Deficits: Inferred from a lack of cohesive devices, tangential storytelling, or a failure to provide context for a listener, making narratives hard to follow.Semantic and Lexical Differences: This can manifest as an overly formal or pedantic speaking style (“professorial”), or conversely, as a limited and concrete vocabulary. A strong focus on a specific topic can be detected through repetitive lexical choices.Pronoun Reversal: A historically noted but variable feature where the individual might refer to themselves as “you” or by their own name.

#### 1.3.4. Movement Analysis

This modality quantifies the neuromotor signs and gestural abnormalities often observed in ASD. Key discriminators include the following:Atypical Gait and Posture: Quantifiable as increased body sway, unusual trunk posture, or asymmetrical, irregular gait.Impoverished or Atypical Gestures: A reduced rate and range of communicative gestures (e.g., pointing and waving) can be measured spatially and temporally. Gestures that are present may be poorly integrated with speech or appear stiff and mechanical.Motor Incoordination: Measured as increased movement irregularity, high variability, and reduced smoothness in reaching and other goal-directed movements.

#### 1.3.5. Activity Recognition

This modality focuses on identifying and classifying patterns of gross motor activity or behavioural episodes over time, often to capture restricted and repetitive behaviours (RRB) or atypical patterns of social interaction. This differs from movement analysis, which extracts fine-grained spatial–temporal details from skeletal or kinematic data. Its discriminative power lies in detecting the following:Motor Stereotypies: The ability to automatically classify whole-body repetitive behaviours such as body rocking, hand flapping, or spinning.Atypical Activity Patterns: Recognizing periods of unusually high or low activity levels, or a lack of variation in play and exploration.Context-Inappropriate Behaviours: Identifying episodes such as elopement (running off) or meltdowns characterized by intense motor agitation.

#### 1.3.6. Facial Gesture Analysis

This modality focuses on the production and analysis of facial expressions. It provides a quantitative basis for the DSM-5 observation of “deficits in nonverbal communicative behaviors”, specifically “abnormalities in facial expression”. Key discriminators include the following:Reduced Facial Expressivity: A lower frequency and intensity of facial Action Unit (AU) activation, leading to a “flat affect” or neutral face, even in emotionally charged situations.Atypical Expression Dynamics: Facial expressions may be fleeting, slow to develop, or poorly timed with social cues and speech.Incoherent Expression: Incongruence between different parts of the face (e.g., a smiling mouth without the accompanying eye creases of a genuine “Duchenne smile”).

#### 1.3.7. Visual Attention

This modality measures how individuals allocate their visual attention in social and non-social contexts. The most robust discriminators are outlined as follows:Reduced Attention to Social Stimuli: Quantifiably less time spent fixating on the eyes and faces of others in images, videos, or real-life interactions.Atypical Scanpaths: Visual exploration of a scene that is more disorganized, with more fixations on background objects and fewer on socially relevant information.Impaired Joint Attention: A failure to spontaneously follow another person’s gaze or pointing gesture, a key early social milestone.

#### 1.3.8. Multimodal Approaches

This approach recognizes that ASD is a whole-body condition and seeks to capture its interconnected behavioral signature by fusing data streams. Its discriminative power comes from the following:Capturing Behavioral Incongruence: For example, detecting when a flat vocal prosody (Voice Biomarker) co-occurs with a lack of facial expressivity (Facial Gesture Analysis) during a supposedly happy conversation.Compensatory Analysis: Identifying when a strength in one modality (e.g., complex linguistic content) masks a deficit in another (e.g., poor interactional dynamics).Holistic Phenotyping: Creating a composite profile that more accurately reflects the complex, multi-faceted nature of ASD than any single modality can. A model might fuse visual attention (to eyes), vocal prosody, and head movement to generate a more robust estimate of social engagement.

## 2. Materials and Methods

### 2.1. Identification

This review is organized into several key sections, according to the Preferred Reporting Items for Systematic Reviews and Meta-Analyses extension for Scoping Reviews (PRISMA-ScR) framework [9].

To ensure comprehensiveness and transparency, a structured search strategy was implemented. The literature search was conducted across the following five major bibliographic databases: PubMed, IEEE Xplore, ScienceDirect, Scopus, and ArXiv. These databases were selected to cover complementary domains relevant to the review, including biomedical sciences, psychology, neuroscience, engineering, and computational research, as follows:PubMed was searched for its extensive coverage of biomedical and healthcare research, particularly studies on neurodevelopmental disorders and clinical applications of AI.IEEE Xplore was consulted for research on machine learning algorithms, signal processing, and computational models applied to ASD detection.ScienceDirect provided access to literature spanning psychology, neuroscience, and computational approaches in healthcare.Scopus and ArXiv ensured multidisciplinary coverage, including peer-reviewed and preprint studies across medical, engineering, and behavioral sciences.

In addition to these databases, publisher platforms such as SpringerLink and MDPI were also consulted to capture potentially relevant records not indexed elsewhere.

To identify the relevant literature for each modality outlined in Section 1.3, a structured search strategy was implemented across selected databases and platforms. The search utilized targeted Boolean queries combining the following key terms: *“Detection of neurodevelopmental diseases”*, *“Autism”*, *“Multimodal data”*, *“AI in health”*, *“Machine learning”*, *“Deep learning”*, *“Convolutional neural networks”*, *“Video analysis”*, *“Behavioral data”*, *“Voice acoustic”*, *“Voice biomarker”*, *“Body movement”*, *“Movement analysis”*, *“Motor skills”*, *“Facial gesture analysis”*, *“Visual attention”*, *“Activity recognition”*, *“Conversational Interactional Dynamics”*, *“Speech recognition”*, *“Speech processing”*, and *“Language analysis”*.

The use of multiple bibliographic databases and supplementary publisher platforms, combined with advanced filtering and exporting tools, ensured a systematic and reproducible literature search process.

### 2.2. Selection of Sources of Evidence

The selection of sources of evidence or the screening phase was guided by the following four exclusion criteria to ensure the relevance and quality of the collected articles:Elimination of duplicates to maintain a unique and non-redundant dataset;Inclusion of English-language publications to ensure accessibility and consistency in analysis;A 10-year publication range was chosen to encompass the rapid advances and emerging trends in AI-driven methodologies.To ensure a focused and high-quality analysis, only research articles were selected for this screening phase.

### 2.3. Eligibility

To further refine the selection, an additional level of duplicate removal was performed using Zotero bibliographic reference management software. Research considered eligible for inclusion was identified based on the following criteria:Full-text availability, ensuring comprehensive access to research findings.Peer-reviewed journal publications, guaranteeing scientific rigor and credibility.

### 2.4. Studies Included in Scoping

Only studies that specifically applied AI techniques to the diagnosis of ASD, with sufficient methodological detail and experimental validation, were included in the final selection (Articles Included) to ensure methodological robustness and relevance to research objectives. In summary, the search process and the selected articles, according to the PRISMA methodology, are presented in Figure 1.

## 3. Results

### 3.1. Overview of Included Studies

Databases such as arXiv, IEEE Xplore, MDPI, PubMed, ScienceDirect, Scopus, Springer, and others contributed varying quantities of articles at each stage of the review process. Figure 2 presents the total number of research articles that were identified, screened, considered eligible, and ultimately included in the scoping review, organized by source database. The visualization highlights the flow and reduction of articles through the selection stages, providing insight into the relevance and volume of the literature retrieved from each platform.

Also, Figure 3 illustrates the quantity of research studies yielding tangible outcomes across different modalities. Notably, autism detection through visual attention has garnered significant research interest. Conversely, leveraging voice biomarkers emerges as both the most straightforward to implement and the most effective approach, as shown in Table 1, warranting further investigation for potential clinical applications.

Within the broad spectrum of AI approaches to ASD detection, algorithms leveraging voice processing can be typically classified into the following three categories: voice biomarkers, interactional dynamics, and linguistic content analysis. The following subsections first examine these three categories in detail, outlining the specific AI methods applied in each, before introducing additional approaches that are primarily vision-based, such as movement analysis, activity recognition, facial gesture analysis, visual attention, and multimodal approaches. For each category, we also highlight key methodological challenges and potential opportunities identified in the literature.

### 3.2. Voice Biomarkers (Acoustic/Prosodic Features)

Recently, voice biomarkers have gained considerable attention for their potential to reveal clinically relevant patterns in speech. In the literature, alternative terms are also used for this category, including “voice bio-markers”, “vocal biomarkers”, and “acoustic biomarkers”. Voice biomarkers focus on quantifiable acoustic and prosodic characteristics, such as pitch, jitter, shimmer, formants, and spectral energy, capturing measurable deviations in vocal production that may indicate ASD, independent of language content.

In this review, 13 research articles were identified under this modality. Two representative works of Vacca et al. (2024) [2] and Briend et al. (2023) [10] are discussed in detail below, followed by a brief synthesis of the remaining studies.

#### 3.2.1. Data Preprocessing Approaches

In the study by Vacca et al. (2024) [2], preprocessing involved manually clipping audio signals to remove silent segments, followed by noise filtering in Audacity. Noise reduction (6 dB), sensitivity, and frequency damping parameters were manually tuned to minimize distortion. Signals were then pre-emphasized using a high-pass FIR filter and normalized to standardize amplitude variations.

Briend et al. (2023) [10] recorded speech samples from 108 children, 38 with ASD, 24 typically developing (TD), and 46 with atypical development using a non-word repetition task. Recordings were segmented in Praat with visually marked boundaries, and low-quality or outlier samples were excluded to ensure clean input.

Hu et al. (2024) [11] used data from the Caltech audio dataset, which contains dialogues from the Autism Diagnostic Observation Schedule (ADOS). Preprocessing included extracting 40 speech features spanning frequency, zero-crossing rate, energy, spectral characteristics, and Mel-Frequency Cepstral Coefficients (MFCCs) to capture subtle differences between ASD and non-ASD speech.

Similarly, Keerthana Sai et al. (2024) [12] developed the Children’s ASD Speech Corpus (CASD-SC) and applied short-time Fourier transform (STFT) to convert raw speech into time–frequency representations for convolutional neural network (CNN) analysis. Both raw and augmented data were used to improve robustness.

Other studies adopted varied preprocessing strategies as follows: Mohanta and Mittal (2022) [13] used audio segmentation and normalization; Jayasree and Shia (2021) [14] applied denoising and feature scaling; Asgari and Chen (2021) [15] combined segmentation with normalization to address recording variability; Godel et al. (2023) [16] paired normalization with manual annotation; and Guo et al. (2022) [17] applied tone segmentation for Mandarin speech.

#### 3.2.2. Feature Extraction

Vacca et al. (2024) [2] extracted 36 features as follows: formant frequencies (F1–F5), dominant frequencies (FD1, FD2), pitch (F0), jitter, shimmer, energy, zero-crossing rate (ZCR), MFCCs, and LPCCs, illustrated in Figure 4a,c. Framing (25 ms, 10 ms overlap) and Hamming windows reduced spectral distortion as shown in Figure 4b, Briend et al. [10] focused on nine acoustic parameters, F0, F1–F4, formant dispersion, HNR, jitter, shimmer, plus skewness, and kurtosis, to quantify intra-individual variability.

Hu et al. (2024) [11] focus on extracting a diverse array of speech features that encapsulate various dimensions of communication. More specifically, they propose grouping the speech features for ASD assessment into eight categories as follows: intonation (pitch-related measures such as fundamental frequency and MFCCs), volume (energy and its variation), rhythm (signal zero-crossing rates), rate (speech speed and density), pauses (frequency and balance between speaking and silence), spectral features (describing the frequency distribution and its changes), chroma (pitch class profiles), and duration (speaking time with or without pauses). These features capture prosody, articulation, and temporal patterns that can distinguish ASD from typical speech. Figure 5 summarizes the studied speech feature categories. In contrast, Keerthana Sai et al. (2024) [12] specifically utilized STFT to create log spectrogram representations of the speech signals, which served as inputs to their layered CNN model. This method not only captured the temporal and spectral dynamics of speech but also allowed for a detailed analysis of the differences in speech patterns between children with ASD and their typically developing peers.

Other studies explored diverse acoustic descriptors, including the following:Mohanta & Mittal (2022) [13]: formants, MFCCs, and jitter.Jayasree & Shia (2021) [14]: MFCCs, jitter, and shimmer.Asgari & Chen (2021) [15]: pitch, jitter, shimmer, MFCCs, and prosodic cues.Godel et al. (2023) [16] and Lau et al. (2022) [18]: intonation, rhythm, stress, and duration.Lee et al. (2020) [19]: prosody, MFCC, and spectral measures.Guo et al. (2022) [17]: jitter, shimmer, HNR, and MFCCs.Li et al. (2019) [20]: acoustic features using OpenSMILE and a spectrogram.Rosales-Pérez et al. (2019) [21]: acoustic features acquisition from the Mel Frequency Cepstral Coefficient and the Linear Predictive Coding from signal processing.Chi et al. (2020) [22]: audio features (including Mel-frequency cepstral coefficients), spectrograms, and speech representations (wav2vec).

#### 3.2.3. AI Techniques Used

Vacca et al. [2] tested random forests (RF), support vector machine (SVM), Logistic Regression, and Naïve Bayes classifiers, using *t*-test–based feature selection. Briend et al. [10] applied k-means clustering guided by ROC curve supervision with Monte Carlo cross-validation (500 iterations). Hu et al. (2024) [11] evaluated both classification and regression models for distinguishing ASD from non-ASD cases, achieving a classification accuracy of 87.75%. Keerthana Sai et al. (2024) [12] proposed a customized STFT-layered CNN for raw speech analysis, incorporating data augmentation to improve robustness.

Other works span both classical and deep learning as follows:Mohanta & Mittal (2022) [13]: GMM, SVM, RF, KNN, and PNN.Jayasree & Shia (2021) [14]: FFNN.Asgari & Chen (2021) [15]: SVM and RF.Godel et al. (2023) [16]: statistical models and ML classifiers.Lau et al. (2022) [18]: SVM.Lee et al. (2023) [19]: SVM, RF, and CNN.Guo et al. (2022) [17]: RF.Li et al. (2022) [20]: SVM and CNN.Chi et al. (2022) [22]: RF, CNN, and Transformer-based ASR model.

#### 3.2.4. Performance, Constraints, and Challenges

Vacca et al. [2] reported high accuracy for SVM and RF (98.82%), with precision and recall above 97% for ASD classification. However, the small dataset (84 subjects) and high feature-to-sample ratio may limit generalizability.

Briend et al. [10] achieved 91% accuracy when distinguishing ASD from neurotypical children, and 85% when compared to a heterogeneous control group, identifying shimmer and F1 as key discriminators. Limitations included gender imbalance (ASD group predominantly male) and the need for validation in younger cohorts.

Both studies highlighted the following common challenges: limited multilingual or multi-age datasets, the impact of comorbidities (e.g., Attention-Deficit/Hyperactivity Disorder), and the absence of longitudinal data. Future research should focus on larger and more diverse samples, broader linguistic contexts, and the exploration of cry-based biomarkers for early ASD detection.

Hu et al. (2024) [11] achieved 87.75% classification accuracy using machine learning models, demonstrating not only improved detection performance but also the potential for more objective characterization of speech-related behaviors to support personalized interventions. Nonetheless, validation on larger, more heterogeneous datasets remains necessary.

Similarly, Keerthana Sai et al. (2024) [12] proposed a customized STFT-layered CNN, reaching 99.1% accuracy with data augmentation. Despite the strong results, they acknowledged limited generalizability, particularly in low- and middle-income countries where ASD research is scarce.

Collectively, these studies underscore the promise of machine and deep learning for enhancing ASD diagnosis, while also pointing to the need for continued efforts to address dataset diversity, population variability, and the complex nature of ASD presentations to ensure the development of more robust and accessible diagnostic tools.

#### 3.2.5. Comparative Overview of Top Studies

Table 2 presents a comparative analysis of 10 high-accuracy research works (2015–present) on ASD detection through voice biomarkers using AI techniques. The table summarizes each study in terms of dataset composition, participant age range, preprocessing methods, extracted feature sets, and applied AI models.

### 3.3. Conversational/Interactional Dynamics

Within the broader taxonomy of voice-based ASD markers comprising voice biomarkers, linguistic content, and interactional dynamics, this subsection focuses on AI approaches that analyze how people speak with others rather than what they say. The emphasis is on timing- and prosody-related measures that characterize interactional behavior, such as turn-taking gaps and overlaps, response latency, pause structure, speaking rate coordination, and prosodic entrainment (synchrony of F0, intensity, and tempo) during dialogue.

These interactional features capture pragmatic and social-communication mechanisms that are central to ASD and complement both acoustic biomarkers derived from isolated speech and linguistic content measures from other modalities.

Commonly used interactional metrics include the following:Turn-gap duration and variability.Overlap proportion and pause ratios.Prosodic entrainment (synchrony of F0, intensity, and speech rate) across turns.Conversational balance (reciprocity, turn counts, and initiations vs. responses).Blockwise dynamics of pitch and intensity during dialogue.

#### 3.3.1. Data Preprocessing Approaches

Typical preprocessing pipelines for interactional speech analysis begin with longform audio recorded in naturalistic or semi-structured settings, such as free conversation, story retell tasks, Autism Diagnostic Observation Schedule (ADOS) interviews, or daylong recordings. Processing generally involves the following:Speaker diarization and voice activity detection (VAD) to separate interlocutors and silence.Optional automatic speech recognition (ASR) for transcript alignment when lexical information is required.Time alignment of interlocutor streams to compute dyadic timing measures (e.g., turn boundaries, gap and overlap durations) and prosodic entrainment metrics (cross-correlations of F0, intensity, and speaking rate).

In large-scale daylong studies, Language ENvironment Analysis (LENA)-based pipelines provide automated counts of adult words, child vocalizations, and conversational turns. In laboratory dialogues, higher-resolution diarization and prosodic tracking are typically employed to capture fine-grained interactional features.

Two representative studies illustrate this workflow. Ochi et al. (2019) [23] recorded natural conversations between adults with high-functioning ASD and neurotypical controls. Audio was segmented via speaker diarization, pauses were detected automatically, and turn boundaries were labeled for statistical analysis.

Lehnert-LeHouillier et al. (2020) [24] segmented child and adolescent dialogues into speaker turns, extracted prosodic contours (F0, intensity, and speech rate) for each turn using Praat, and aligned them across speakers to compute entrainment indices.

Other works have adapted this general approach to various contexts as follows: Lau et al. (2022) [18] tracked prosody to study entrainment in individuals with ASD, their relatives, and controls; Wehrle et al. (2023) [25] analyzed free-dialogue turn-gap distributions; Yang et al. (2023) diarized therapy sessions to assess conversation quality; and Chowdhury et al. (2024) [26] segmented examiner–child ADOS dialogues for timing and prosody features.

#### 3.3.2. Feature Extraction

Interactional feature sets in ASD-related speech analysis commonly include the following:Turn-taking timing: Distribution of silence gaps between turns; overlap proportion/frequency; mean/variance of response latency; and overall pause ratio.Prosodic synchrony (entrainment): Correlation or convergence of F0, intensity, and speaking rate across turns, computed as global or local entrainment indices.Participation balance: Relative number and length of turns per speaker; reciprocity of exchanges; and ratio of initiations to responses.Blockwise dynamics: Short-window (e.g., 3–5 s) changes in intensity and pitch aligned across speakers to capture interactional synchrony.

These metrics are typically aggregated at the dyad level and then used as inputs to downstream classification or regression models (e.g., group discrimination or symptom-severity prediction).

#### 3.3.3. AI Techniques Used

Supervised machine learning classifiers such as SVM, RF, and logistic regression (LR), as well as generalized linear models, are commonly used to differentiate ASD from typically developing (TD) individuals based on turn-level timing and prosodic entrainment features. Some studies employ sequence models (e.g., hidden Markov models [HMMs], recurrent neural networks [RNNs]) to capture temporal dependencies across multiple turns, while others extract graph- or correlation-based synchrony metrics as input features.

In large-scale, naturalistic corpora, such as daylong audio from LENA systems, models typically use automated conversational-turn counts and timing distributions, often correlating these features with developmental outcomes rather than performing direct classification. In contrast, controlled laboratory tasks allow for higher-resolution measurements of prosodic entrainment, enabling more detailed classification or regression analyses.

Representative examples include Ochi et al. [23], who applied mixed-effects regression to compare pause and timing profiles across groups; Lehnert-LeHouillier et al. [24], who computed global and local entrainment indices and analyzed them using mixed-effects models; and Yang et al. [27] and Chowdhury et al. [26], who used SVM and RF classifiers on timing and prosody features for ASD vs. TD discrimination. Recent work has also explored predicting perceived conversation quality as an objective proxy measure for intervention monitoring.

#### 3.3.4. Performance, Constraints, and Challenges

Evidence from interactional speech studies suggests that high-functioning autistic participants may exhibit longer turn gaps, higher pause ratios, reduced reciprocity, and diminished synchrony in intensity changes compared to neurotypical peers, particularly in task-oriented dialogues. However, results are not consistent, and some adult-dyad studies report no significant differences in turn-timing distributions. These mixed findings highlight the influence of task design, participant age, language, and interaction context.

Methodological choices also shape results. Ecological data from daylong LENA recordings provide scalable estimates of conversational turns but can be limited by diarization accuracy and lack of contextual detail. Laboratory-controlled dialogues allow for the precise measurement of timing and entrainment metrics but may sacrifice ecological validity.

Open challenges include standardizing interaction tasks, ensuring cross-linguistic robustness, controlling for language ability and comorbidities, and integrating timing and entrainment features with prosody-only and content-based models to improve generalizability.

#### 3.3.5. Comparative Overview of Representative Studies

Table 3 provides a comparative analysis of key studies that investigate conversational and interactional dynamics in ASD using AI-based approaches. The table summarizes each study in terms of dataset size, participant age range, preprocessing procedures, extracted interactional and prosodic features, and the AI models employed along with their reported performance. This overview highlights how different methodological and computational strategies have been applied to capture timing-, rhythm-, and synchrony-related markers of social communication in both controlled and naturalistic conversational settings.

### 3.4. Language- and Content-Level Analysis

This modality focuses on the analysis of lexical, syntactic, semantic, and narrative features derived from spoken or written language. Unlike voice biomarkers, the emphasis here is on what is said, rather than on acoustic characteristics. Data may be sourced directly from transcripts or obtained via automatic speech recognition (ASR) from recorded speech.

#### 3.4.1. Methods of Data Preprocessing

Text-based pipelines typically involve tokenization, normalization, stopword removal, and lemmatization. For ASR-based pipelines, raw speech is first converted to text before applying standard natural language processing (NLP) preprocessing steps.

Representative works include Mukherjee et al. (2023a) [28], who cleaned and tokenized 1200 child-produced text samples, and Rubio-Martín et al. (2024) [29], who normalized and lemmatized 2500 text samples for Term Frequency-Inverse Document Frequency (TF-IDF) and embedding extraction. Deng et al. (2024) [30] and Angelopoulou et al. (2024) [31] focused on narrative speech transcription and cleaning, while Hansen et al. (2023) [32] combined acoustic prosodic features with lexical and syntactic markers through separate models and ensemble integration.

#### 3.4.2. Feature Extraction

Feature categories include the following:**Lexical:** Vocabulary richness, type-token ratio, and word frequency.**Syntactic:** Sentence length, parse tree depth, and part-of-speech tags.**Semantic:** Embeddings from BERT, GPT, or word2vec.**Narrative:** Cohesion, coherence, and temporal structure.

#### 3.4.3. AI Techniques Used

Supervised classifiers such as SVM and RF are common, along with deep learning models like CNN, long short-term memory (LSTM), BiLSTM, and transformer-based architectures (e.g., BERT and GPT). Multimodal models may incorporate language features alongside prosodic or visual inputs.

#### 3.4.4. Performance, Constraints, and Challenges

Reported accuracies range from 88% to 92%, with transformer-based models often achieving the best performance. Limitations include small, homogeneous datasets; limited cross-linguistic validation; and potential bias from ASR transcription errors.

#### 3.4.5. Comparison of Autism Language Analysis Classification Studies

Table 4 presents a comparative analysis that apply AI-based language and content-level analysis for ASD detection. Each study is summarized in terms of dataset size, participant age range, text or speech source, preprocessing procedures, extracted lexical, syntactic, and semantic features, as well as the AI models used and their corresponding performance. This overview highlights the evolution from traditional frequency-based and syntactic metrics to modern embedding-based representations leveraging transformer architectures.

### 3.5. Movement Analysis

Movement Analysis refers to studies that extract and analyze fine-grained, sub-motor kinematic features, such as joint angles, limb velocity, tremor frequency, or gait parameters. The primary unit of analysis is a quantitative kinematic variable, often captured via high-resolution video (pose estimation) or specialized motion capture systems. The focus is on the constituent parts of a movement, which may not be explicitly labeled as a named behavior.

This is distinct from Activity Recognition, which primarily involves identifying broad categories of actions such as walking, running, or resting using wearable sensors like accelerometers, gyroscopes, or magnetometers. While both areas relate to physical motion, activity recognition from wearables does not capture the detailed spatial configuration or fine-grained temporal evolution of body movements.

All studies included in Section 3.5, i.e., Airaksinen et al. [36], Rad et al. [37], Zahan et al. [38], Serna-Aguilera et al. [39], Zhao et al. [40], Caruso et al. [41], Alcañiz et al. [42], Bruschetta et al. [43], Sadouk et al. [44], Großekathöfer et al. [45], Jin et al. [46], Doi et al. [47], Wedyan et al. [48], Tunçgenç et al. [49], Luongo et al. [50], Martin et al. [51], Kojovic et al. [52], Mohd et al. [53], Li et al. [54], Rad et al. [55], Altozano et al. [56], Emanuele et al. [57], Lu et al. [58], Sun et al. [59], Ullah et al. [60], Georgescu et al. [61], Siddiqui et al. [62], Ganai et al. [63], Zhao et al. [64], Li et al. [65], and Al-Jubouri et al. [66], are dedicated to ASD detection and use video, depth, or motion capture data to extract detailed motor features.

#### 3.5.1. Data Preprocessing Approaches

Data preprocessing is essential for ensuring the quality, consistency, and comparability of movement data used in ASD detection. Raw movement recordings from video, depth sensing, motion capture, or wearable inertial sensors often contain noise, missing frames, occlusions, or background distractions that must be addressed before analysis. Effective preprocessing standardizes inputs, reduces irrelevant variation, and enhances the representation of movement features for downstream models.

Data cleaning and filtering typically include spatial and temporal smoothing to remove jitter in skeletal joint tracking, interpolation to recover missing joints caused by self-occlusion or sensor dropout, and outlier detection to correct implausible positions, velocities, or accelerations.

Spatial and temporal normalization involves aligning skeletal coordinates or sensor axes to a consistent reference frame (for example, pelvis-centered, camera-centered, or gravity-aligned) to enable cross-subject comparisons, adjusting sequence lengths through dynamic time warping or padding/truncation, and resampling videos or sensor time series to a uniform rate for temporal consistency.

Feature representation preparation converts raw signals into formats suitable for analysis, such as joint coordinates, angles, or motion vectors for skeleton-based encoding; spatial heatmaps of keypoints for convolutional models; optical flow representations to capture fine-grained motion between frames; or statistical/temporal features extracted from accelerometer and gyroscope streams.

Augmentation strategies enhance model robustness through random scaling, rotation, or mirroring skeletons; time-shifting or sub-sequence extraction to account for natural variability; and controlled noise injection to simulate real-world acquisition conditions. For wearable sensor data, this may also include synthetic perturbations to orientation or magnitude to model device placement variation.

Modality-specific preprocessing addresses the unique needs of each acquisition method. For video-based analysis, background subtraction and person segmentation isolate the subject from environmental noise. For depth data, hole filling and surface smoothing improve skeletal estimation accuracy. For motion capture, marker gap filling and trajectory smoothing preserve the continuity of motion signals. For wearable IMU data, gravity compensation, signal filtering (low-pass and band-pass), and calibration across devices are critical in ensuring accuracy.

Several studies illustrate these approaches.

Video/Depth-based: Al-Jubouri et al. (2020) [66] processed Kinect v2 depth videos to isolate a single gait cycle per child using a distance-between-feet heuristic, imputed missing data via predictive mean matching (MICE), anonymized faces with Gaussian blurring, and applied extensive augmentation to address class imbalance. Li et al. (2023) [65] curated therapy-session videos, removed short/noisy clips, and generated privacy-preserving streams including optical flow, 2D/3D skeletons, and Lucas–Kanade flow; these were paired with clinical scores for benchmarking. Other works such as Zhang et al. (2021) [67] and Serna-Aguilera et al. (2024) [39] used context-specific preprocessing, including frame curation, normalization, and skeleton sequence filtering.

By integrating modality-specific preprocessing strategies, these studies ensure that both video-based kinematics and wearable sensor time series are transformed into robust, comparable inputs suitable for ASD-related movement analysis.

#### 3.5.2. Feature Extraction

Feature extraction in movement analysis for ASD detection involves transforming raw spatiotemporal data from video, depth sensors, motion capture systems, or wearable inertial measurement units (IMUs) into numerical descriptors that capture relevant aspects of motor patterns. The objective is to derive discriminative representations of gait, posture, gestures, and other movement dynamics that may signal atypical motor behavior.

A common approach is the extraction of skeletal and kinematic features, where 2D or 3D joint coordinates serve as the basis for computing joint angles, angular velocities, accelerations, and displacement trajectories. From these, higher-level measures such as stride length, inter-limb coordination, joint-to-ground distances, range of motion, and symmetry indices are derived. For example, Li et al. (2023) [65] and Al-Jubouri et al. (2020) [66] used skeletal tracking from depth cameras to quantify gait and coordination patterns, with Al-Jubouri et al. engineering more than 1200 3D gait features subsequently reduced via PCA.

For wearable sensor–based studies, raw accelerometer and gyroscope signals are often decomposed into temporal, statistical, and frequency-domain features. These include step counts, stride times, variability measures, power spectral density in specific gait frequency bands, and inter-axis correlations.

Temporal dynamics and motion patterns capture the evolution of movement over time through statistical summaries (e.g., mean, variance, and range) or sequential modeling techniques. Examples include computing temporal correlations between key joints to assess coordination (Serna-Aguilera et al., 2024 [39]), analyzing velocity profiles to measure motion smoothness or abruptness (Sadek et al., 2023 [68]), and detecting bouts of stereotypies such as head-banging or arm-flapping based on rhythmicity, repetition rate, bout duration, and spatial amplitude.

Spatial–temporal descriptors integrate positional and timing information, often using 3D joint trajectory tensors or graph-based models that represent joints as nodes and anatomical connections as edges. Spatial–temporal graph convolutional networks (ST-GCNs), such as that of Altozano et al. (2024) [56], automatically learn joint interaction features for ASD classification.

When full skeletal extraction is unreliable, appearance and motion cues are used. These include optical flow fields, motion energy images, or pose heatmaps that encode pixel-level movement information. For example, Ganai et al. (2025) [63] combined optical flow maps with pose heatmaps to integrate appearance-based and structural movement cues. MMASD (Li et al., 2023) [65] distributes features across privacy-preserving streams, including flow and skeleton data, to enable standardized benchmarking.

Finally, higher-level motor behavior indicators such as movement variability indices, symmetry scores, and frequency-domain representations help capture subtle deviations in rhythm and coordination that are not evident from raw positions alone. Infant and early-screening studies (Doi et al., 2022 [47]; Bruschetta et al., 2025 [43]) often use markerless body-tracking to extract spontaneous movement statistics relevant to developmental assessment.

By combining skeletal geometry, wearable-sensor kinematics, temporal dynamics, spatial–temporal structure, appearance cues, and higher-order behavioral measures, these approaches create comprehensive movement representations that enhance the sensitivity and robustness of ASD detection models.

#### 3.5.3. AI Techniques Used

Studies in the Movement Analysis category apply a variety of AI approaches to model the spatiotemporal patterns of motor behavior and detect atypical movements associated with ASD. Methods range from conventional machine learning on handcrafted features to deep learning architectures that learn spatiotemporal representations directly from raw or minimally processed data.

Classical machine learning on handcrafted features remains common for structured kinematic inputs such as gait, posture, and wearable sensor measures. These approaches use manually engineered descriptors, including joint displacements, velocities, accelerations, gait parameters, postural stability metrics, and frequency-domain features from accelerometer or gyroscope signals. The extracted features are then classified using algorithms such as multilayer perceptrons (MLP), SVM, or RF. For example, Al-Jubouri et al. (2020) [66] trained an MLP on Kinect-derived gait metrics.

Convolutional neural networks (CNNs) are effective when movement data are represented as pose heatmaps, motion images, or optical flow fields. Ganai et al. (2025) [63] used CNNs to integrate structural cues from pose heatmaps with motion information from optical flow, which enabled the joint modeling of appearance and movement characteristics.

Recurrent neural networks (RNNs), particularly LSTM models, are applied to sequences of skeletal joint coordinates, wearable-sensor time series, or frame-level features to capture temporal dependencies in movement. Li et al. (2023) [65] used LSTMs to model changes in gait and posture over time.

Graph-based neural networks, such as spatial–temporal graph convolutional networks (ST-GCNs), treat joints as nodes and skeletal connections as edges. These models enable simultaneous modeling of spatial relationships and temporal evolution. Altozano et al. [56] implemented an ST-GCN to learn movement representations directly from skeleton graphs.

Hybrid and multi-stream architectures combine different feature types or modalities, such as skeletal trajectories, velocity-based features, wearable-derived kinematics, and appearance cues, in parallel network streams that are fused at later stages. Serna-Aguilera et al. (2024) [39] adopted this strategy to improve classification accuracy.

Recent trends indicate a shift toward deep learning approaches, particularly graph-based and multi-stream models, because they support end-to-end learning of spatiotemporal features and reduce reliance on handcrafted inputs. Classical machine learning remains competitive for highly structured kinematic or wearable data. Curated datasets, such as MMASD (Li et al., 2023 [65]), have become important for establishing reproducible benchmarks and enabling fair model comparisons.

#### 3.5.4. Performance, Constraints, and Challenges

Reported performance varies with task design and dataset realism. Al-Jubouri et al. reported a high classification accuracy of 95% with their MLP model. Stereotypy detection from curated videos typically achieves high accuracy, as in head-banging or arm-flapping classification [68,69]. In contrast, in-the-wild recordings from home or clinical environments introduce challenges such as occlusion, variable viewpoints, and motion-tracking instability [39,70]. Gait and posture models show promise for early screening [54,63], although they require harmonized protocols and age-matched control groups to ensure comparability. Markerless infant movement analyses demonstrate predictive value at 4 to 18 months [43,47], but they also require standardized capture procedures and cross-site validation. Privacy-preserving representations, such as optical flow and skeleton-based encodings, help reduce identifiability risks while retaining essential movement information [65].

#### 3.5.5. Comparative Overview of Representative Studies

This section presents in Table 5 a comparative summary of key studies in movement analysis for ASD detection, with a focus on their data sources, feature extraction strategies, AI techniques, and reported performance. Only works that meet the defined criteria for movement analysis are included. These criteria require the use of video, depth sensing, or motion capture to record fine-grained spatial and temporal motor patterns. Studies that rely solely on wearable devices to measure general activity levels are excluded, as such works are classified under the activity recognition category. The table illustrates the diversity of sensor modalities, processing pipelines, and model architectures, reflecting both the breadth of approaches explored in the field and the growing preference for deep learning techniques capable of learning movement representations directly from raw spatial and temporal data.

### 3.6. Activity Recognition

Activity recognition in the context of ASD detection typically relies on broader behavioral patterns inferred from wearable sensors (e.g., accelerometers and gyroscopes) or from video sequences analyzed at a coarser level. This approach is particularly effective for continuous monitoring and for detecting behaviors that manifest in longer temporal windows, such as stereotypies, meltdowns, or reduced social engagement.

So, it refers to studies aimed at classifying pre-defined, macro-level behavioral episodes or states, such as ‘hand-flapping’, ‘body rocking’, or ‘avoidance behavior’. While these studies may use kinematic data (e.g., from pose estimation), the primary output is a behavioral label, not a continuous kinematic variable. The movement data serves as an input feature for recognizing the larger, recognizable activity.

References pertaining to the activity recognition modality are listed in the References section, covering entries [30] through [72], as follows: Singh et al. [73], Lakkapragada et al. [74], Cook et al. [70], Sadek et al. [68,69], Asmetha et al. [75], Stevens et al. [76], Jayaprakash et al. [77], Gardner-hoag et al. [78], Wei et al. [71], Gomez-donoso et al. [79], Zhang et al. [80], Zhao et al. [81], Li et al. [82], Deng et al. [30], and Jin et al. [46].

#### 3.6.1. Data Preprocessing Approaches

Preprocessing in activity recognition depends strongly on the data modality. In Deng et al. (2024) [30], audio–visual recordings were synchronized, denoised, and segmented into 1-second chunks to enable frame-level behavior analysis. Video frames were normalized, and composite image representations were generated from sampled frames for downstream feature extraction. Speech transcription and audio captioning were used to convert audio into structured text for multimodal large language model (MLLM) processing.

Li et al. (2024) [82] applied child-specific Faster R-CNN detection for person localization followed by HRNet-based skeleton extraction, retaining 17 keypoints per frame. Skeleton sequences were normalized, and Gaussian smoothing was applied to reduce frame-to-frame noise.

Other works in this category, such as Zhao et al. (2025) [81], Singh et al. (2024) [73], and Alam et al. (2023) [72], used common preprocessing steps including audiovisual alignment, motion segmentation, data augmentation, and normalization. For wearable-based studies, preprocessing often involved resampling sensor signals, noise filtering, and normalization to standard units.

#### 3.6.2. Feature Extraction

Feature extraction strategies vary by modality as follows: In Deng et al. (2024) [30], visual features were derived from CLIP and ImageBind, audio features from Whisper, and multimodal embeddings were obtained at both segment and clip levels. Representations were fused through averaging, maximum pooling, and concatenation strategies before classification.

Li et al. (2024) [82] extracted spatial movement features from skeleton data via a Graph Convolutional Network (GCN) and modeled temporal dynamics with an attention-enhanced LSTM (xLSTM).

Zhao et al. (2025) [81] combined prosodic, gestural, and engagement cues from audiovisual recordings, while Singh et al. (2024) [73] used YOLOv7 for gesture detection and VideoMAE (a transformer-based video model) for temporal motion encoding. Wearable-based works typically computed statistical and frequency-domain descriptors (e.g., mean acceleration, spectral energy, and movement entropy) before classification.

#### 3.6.3. AI Techniques Used

AI methods for activity recognition range from deep learning architectures to classical machine learning as follows: Deng et al. used foundation models (CLIP, ImageBind, and Whisper) integrated into MLLMs such as GPT-4V and LLaVA, with fine-tuning via a post hoc to ad hoc instruction tuning pipeline for improved explainability.

Li et al. [82] implemented a hybrid 2sGCN–AxLSTM model, fusing spatial and temporal features with adaptive weighting optimized via reinforcement learning.

Transformer-based audiovisual models (Zhao et al., 2025) [81], dual-stream self-supervised video transformers (Asmetha & Senthilkumar, 2025 [75]), CNN–LSTM hybrids (Alam et al., 2023 [72]), and ensembles combining SVM, RF, and DNN (Liu & He, 2021 [83]) also appear in this category. Wearable-only works often rely on SVM, RF, gradient boosting, or lightweight deep models for on-device classification.

#### 3.6.4. Performance, Constraints, and Challenges

Performance varies depending on modality and dataset size. The highest reported accuracies (above 90%) were typically achieved by multimodal deep learning systems that combine temporal modeling with feature fusion, as in Zhao et al. (2025) [81] and Singh et al. (2024) [73]. Wearable-only systems achieve competitive but generally lower accuracy, with constraints in capturing nuanced motor behavior.

Challenges across studies include limited dataset sizes, variability in behavior expression across individuals, and reduced interpretability of deep learning models. Clinical applicability requires validation in ecologically valid, real-world settings with diverse populations.

#### 3.6.5. Comparative Overview of Representative Studies

Table 6 presents a comparative overview of representative works in activity recognition for ASD detection, summarizing dataset characteristics, preprocessing pipelines, feature extraction strategies, and AI techniques.

### 3.7. Facial Gesture Analysis

This modality focuses on the analysis of facial gestures in autistic individuals. Typical inputs include RGB video of the face from which frames, facial landmarks, and action units are extracted to quantify socio-emotional signaling. All articles relevant to this modality are cited in the references section, ranging from [4,80,84,85,86,87,88,89,90,91,92,93,94,95,96,97,98,99,100,101,102,103,104,105,106,107,108,109,110,111,112,113].

#### 3.7.1. Data Preprocessing Approaches

In the study by Ruan et al. [86], video recordings from structured ADOS-2 interviews with ASD and typically developing participants were first processed to ensure standardized data input. Each video frame underwent cropping and resizing to a fixed 128 × 128 pixel dimension, with a larger bounding box used around the face to capture any head movement. To address imbalanced group sizes, a variety of data augmentation techniques were applied, such as horizontal flipping, brightness alteration, and histogram equalization.

Almars et al. (2023) [114] employed standard face detection and normalization techniques, including illumination and scale adjustments, along with data augmentation to improve robustness for transfer learning on facial images. Similarly, other face-only pipelines (e.g., Farhat 2024 [113]; Shahzad 2024 [85]; Akter 2021 [99]; ElMouatasim and Ikermane 2023 [91]; Gautam 2023 [109]; Gaddala 2023 [92]; and Vasant 2024 [115]) typically include face cropping, landmark-based alignment, color normalization, class balancing, and stratified train–validation splits.

#### 3.7.2. Feature Extraction

Feature extraction in Ruan et al. [86] was centered on isolating and quantifying micro-expressions brief, involuntary facial movements within selected video intervals. The study employed the Shallow Optical Flow Three-stream CNN (SOFTNet) model to spot candidate micro-expressions, pinpointing their onset and apex within video clips. Each detected segment was then processed using the Micron-BERT framework, which utilizes patch-wise swapping, Patch of Interest (PoI) modules, and Diagonal Micro Attention (DMA) mechanisms. This deep learning approach algorithmically highlighted and encoded subtle facial dynamics by focusing attention on locally important regions and differences between onset and apex.

In contrast, the approach of Farhat et al. [113] relies on deep neural networks, specifically VGG16, which uses convolution and pooling operations to learn critical facial characteristics tied to ASD. Layers transform the input data into discriminative features for classification.

Face-only image studies employed landmark geometry, action-unit statistics, texture descriptors, and deep convolutional embeddings (e.g., VGG and ResNet), sometimes enhanced with attention maps or heatmaps to emphasize discriminative facial regions. Gaze-video studies further quantified fixation density, saccade frequency, and spatial dispersion from eye-region crops or eye-tracking data.

#### 3.7.3. AI Techniques Used

For the classification tasks, Ruan et al. [86] compared the performance of the following three classifiers: multi-layer perceptron (MLP), support vector machine (SVM) with a linear kernel, and ResNet architectures. Final subject-level predictions were derived using a majority voting mechanism across all micro-expression intervals for each participant, if most intervals were classified as ASD, the subject was labeled accordingly.

Farhat et al. [113] undertook a rigorous comparison of multiple classifiers, from deep neural networks (VGG16 and MobileNet) to classical machine learning models (KNN, Random Forest, and Gradient Boosting). The implementation was supported by the Orange toolkit, which provided an automated workflow and advanced visualization capabilities.

Face-image studies often employ transfer learning with CNN backbones such as VGG, ResNet, or DenseNet, sometimes incorporating detector networks (e.g., YOLO variants) for automated facial region extraction. Optimization strategies include fine-tuning, class-balanced loss functions, and simple late-fusion techniques when combining gaze features with facial embeddings [101,103,107].

#### 3.7.4. Performance, Constraints, and Challenges

The linear SVM classifier delivered the highest and most interpretable results in Ruan et al. [86] and achieved a high accuracy of 94.8% overall, with accuracy and F1 scores frequently exceeding 93% across multiple scenarios for certain combinations of assessment topics. The study by Farhat et al. [113] demonstrated that the VGG16 architecture, evaluated using 5-fold cross-validation, was the optimal classifier, attaining a peak validation accuracy of 99% and a testing accuracy of 87%. Although MobileNet delivered comparable performance, its computational efficiency makes it more suitable for deployment in resource-constrained environments. The classifier’s limitation in generalizing across groups highlighted the need for more diverse training data and robust models. Both studies underline challenges such as dataset size, heterogeneity in ASD presentation, and the need for real-time, scalable systems for clinical application.

#### 3.7.5. Comparison of Autism Facial Gesture Analysis Classification Studies

Table 7 offers a comparative summary of the ten most accurate studies (from 2015 to the present) focused on autism detection using facial gesture analysis with AI methods. The studies included in this table are ranked by classification accuracy and provide details on dataset composition, participant age, data preprocessing procedures, feature extraction methods, and advancements in AI models within this evolving research area. Additional studies reviewed are listed as follows: Beary et al. [110], Ji et al. [108], Chen et al. [103], Alam et al. [112], Awaji et al. [101], Alvari et al. [102], Saranya et al. [97], Lu et al. [100], Derbali et al. [96], Alkahtani et al. [93], Sarwani et al. [89], Alhakbani et al. [90], and Muhathir et al. [88].

### 3.8. Visual Attention Analysis

Section 3.8 focuses on methods that quantify gaze allocation and eye movement dynamics as behavioral markers for ASD. These approaches rely on eye-tracking or video-based gaze inference to capture atypical fixation patterns, saccades (rapid eye movements that abruptly shift the point of fixation), and scanpaths in response to social and nonsocial stimuli.

Recently, such approaches have gained attention as simple, non-invasive tools for ASD evaluation. This interest is supported by consistent evidence linking ASD with distinctive visual attention patterns that differ from those observed in typical development.

It is important to distinguish the visual attention category from facial feature analysis. Facial feature analysis primarily investigates the expressive output of the face, such as emotion and micro-expressions. In contrast, visual attention measures perceptual input, which refers to how the eyes move across scenes, interactions, or avatars. Across screen-based, dyadic, and VR paradigms, studies consistently show that gaze distribution and attention to socially salient stimuli provide robust discrimination between ASD and TD groups [116,117].

Fang et al. [118], Xie et al. [3], Chang et al. [119], Pierce et al. [120], Keehn et al. [121], Alcañiz et al. [122], Chong et al. [123], Zhao et al. [124], Alvari et al. [102], Wei et al. [125], Cilia et al. [126], Liaqat et al. [127], Ahmed et al. [128], Antolí et al. [129], Jiang and Zhao [130], Tao and Shyu [131], Mazumdar et al. [132], Minissi et al. [133], Vasant et al. [115], Almadhor et al. [134], Alsaidi et al. [135], Kanhirakadavath and Chandran [136], Sá et al. [137], Ozdemir et al. [138], Varma et al. [139], Thanarajan et al. [140], Islam et al. [141], Solovyova et al. [142], Fernández et al. [143], Shi et al. [144], and Yu et al.[145].

#### 3.8.1. Data Preprocessing Approaches

Most studies on ASD visual attention rely on eye-trackers in screen-based experimental setups, where participants view stimuli on monitors while gaze data are collected. This includes the work of Fang et al. [118], Xie et al. [3], Chang et al. [119], Pierce et al. [120], Keehn et al. [121], Zhao et al. [124], Ahmed et al. [128], Wei et al. [125], Cilia et al. [126], Liaqat et al. [127], Antolí et al. [129], Jiang and Zhao [130], Tao and Shyu [131], Mazumdar et al. [132], Vasant et al. [115], Alsaidi et al. [135], Kanhirakadavath and Chandran [136], Sá et al. [137], Ozdemir et al. [138], Varma et al. [139], Thanarajan et al. [140], Islam et al. [141], and Solovyova et al. [142]. These works primarily investigate gaze allocation, fixation distributions, scanpaths, and saliency in controlled laboratory contexts.

Beyond standard screen-based paradigms, video-based approaches avoid dedicated eye-trackers and instead use cameras to infer gaze or related cues. Chong et al. [123] analyzed video recordings to detect eye contact; Alvari et al. [102] studied micro-expressions as indirect measures of social attention; Fernández et al. [143] developed video-based gaze preference detection; Yu et al.[145] examined atypical gaze in naturalistic videos; and Shi et al. [144] clustered large-scale video-derived gaze data.

More recently, immersive virtual reality (VR) systems with integrated eye-tracking have been employed. Alcañiz et al. [122] used VR for diagnostic simulations, Minissi et al. [133] combined VR with biosignal acquisition, and Almadhor et al. [134] integrated VR gaze with kinematic measurements to capture richer behavioral markers. These approaches offer ecologically valid contexts compared to traditional laboratory setups.

Finally, the review by Minissi et al. [146] synthesized diverse paradigms of machine learning applied to eye-tracking and social visual attention, including both static and video-based methods, in detail.

Preprocessing methods vary depending on the data acquisition technology. For screen-based trackers, common steps include interpolating short gaps, removing outliers, smoothing fixation coordinates, and temporally aligning gaze traces with stimuli. More advanced pipelines also segment gaze into fixations and saccades, map fixations onto Areas of Interest (AOIs), and normalize looking-time features across participants [119,120]. In video-based studies, preprocessing often involves stabilization, synchronization with head pose, and dimensionality reduction techniques such as clustering, PCA, or histogram-based encoding to generate compact gaze descriptors [123,146]. In VR paradigms, gaze vectors must be rendered relative to dynamic 3D environments and avatars, sometimes combined with other biosignals for multimodal integration [122,133]. For egocentric and real-world recordings, calibration and data-quality control are critical, alongside privacy-preserving strategies such as anonymizing or masking faces before analysis.

#### 3.8.2. Feature Extraction

Feature extraction in gaze analysis transforms raw fixations and saccades into discriminative descriptors of visual attention. Approaches range from handcrafted summary metrics to deep learning representations as follows:**Fixation-based metrics:** Proportion of looking time to regions of interest (ROIs), such as eyes, mouth, or faces versus nonsocial content. Pierce et al. [120] quantified attention to “motherese” speech, while Chang et al. [119] measured gaze allocation to social versus nonsocial scenes using percent-looking and silhouette scores. Yang et al. (2024, not in bib yet) extended this approach by modeling transitions between AOIs as graph structures, highlighting reduced face-directed transitions as robust ASD markers.**Scanpath dynamics:** Sequences of fixations and saccades can be represented using methods such as string-edit distances, Markov models, or temporal statistics. Minissi et al. [146] further summarized clustering-based encodings, histogram representations, and dimensionality reduction pipelines for scanpath analysis.**Heatmap encodings:** Converting gaze distributions into spatial maps enables compatibility with convolutional networks. Xie et al. [3] and Wei et al. [125] generated fixation heatmaps for CNN analysis, while Chang et al. [119] used silhouette scores to cluster gaze density patterns.**Deep representations:** Recent work bypasses handcrafted features by learning directly from raw gaze traces. Vasant et al. [115] trained deep neural networks on gaze trajectories and saliency maps, demonstrating higher classification accuracy and scalability in tablet-based screening.**Semantic fusion:** Fang et al. [118] integrated object-level semantics with gaze positions to build hierarchical attention representations. Similar approaches appear in VR paradigms, where gaze is contextualized relative to avatars or interactive objects [122].**Egocentric/real-world cues:** Wearable eye trackers enable measures of naturalistic attention such as eye contact detection [123] or social partner engagement, often requiring privacy-preserving preprocessing.

More advanced pipelines synthesize artificial scanpaths to augment data [127] or embed explainability mechanisms such as influence functions [129] or SHAP-based feature attribution (Almadhor et al., 2025 [134]) to increase interpretability and clinical trust.

#### 3.8.3. AI Techniques Used

AI approaches in visual attention analysis range from classical machine learning on handcrafted gaze features to deep learning architectures operating directly on raw gaze data.

**Classical ML:** Early works applied SVMs, RF, or discriminant analysis to fixation counts, gaze duration ratios, or scanpath statistics [130,138]. Subsequent studies extended these pipelines with clustering, PCA, or histogram-based encodings for dimensionality reduction [136,137,146].**Convolutional models:** CNNs process gaze heatmaps or fixation maps [3,126]. Wei et al. [125] and Mazumdar et al. [132] modeled spatial patterns of visual exploration, while Fernández et al. [143] demonstrated CNN-based gaze preference detection in children with ASD.**Sequential models:** CNN-LSTM hybrids capture temporal dependencies in scanpaths [131,135]. Ahmed et al. [128] proposed an end-to-end pipeline for gaze-driven ASD detection, and Thanarajan et al. [140] used deep optimization frameworks. Recent work also explored involutional convolutions for temporal gaze encoding [141].**Fusion models:** Semantic and multimodal integration strategies enrich gaze-based representations. Fang et al. [118] introduced hierarchical semantic fusion, Alcañiz et al. [122] and Minissi et al. [133] combined gaze with VR contexts and biosignals, and Varma et al. [139] leveraged mobile game-based gaze indicators. Vasant et al. [115] advanced transfer learning on raw gaze trajectories for scalable tablet-based screening.**Explainable ML:** Colonnese et al. [147], Antolí et al. [129], and Almadhor et al. [134] emphasized interpretability, applying techniques such as influence tracing, SHAP-based feature attribution, and task-level explanations to enhance clinical trust. Keehn et al. [121] further highlighted the importance of clinical translation by demonstrating the feasibility of eye-tracking biomarkers for ASD diagnosis in primary care.

Recent works increasingly favor multimodal, deep, and explainable designs to improve generalization, robustness, and clinical adoption.

#### 3.8.4. Performance, Constraints, and Challenges

Performance varies with age group, paradigm, and stimulus design. Screen-based paradigms [119,120] achieve high sensitivity and specificity in toddlers, supporting the potential for early screening. Deep learning gaze heatmap models [3,125,126] report accuracies above 90% on curated datasets, while sequential CNN–LSTM approaches further capture temporal dynamics in scanpaths [128,131,135]. Wearable and VR approaches [122,123,133] demonstrate feasibility in naturalistic or immersive contexts, though robustness to ecological variability and real-world noise remains a challenge. Recent tablet-based pipelines leveraging raw gaze trajectories with transfer learning show promise for scalability in community settings [115].

Several methodological and translational constraints persist. Calibration drift, limited compliance, and motion artifacts are recurrent challenges in infants and toddlers [121,138]. Cross-device variability complicates generalization across eye trackers and consumer hardware [137,146]. Stimulus design can bias gaze distributions; social versus nonsocial contrasts may highlight only certain ASD subgroups rather than the spectrum as a whole [139]. Dataset size and demographic diversity remain limited, raising the concerns of overfitting and reduced external validity [132,136].

Interpretability and clinical integration remain active research priorities. Explainable AI techniques such as influence tracing [147], SHAP values [129], and secure multimodal explanations [134] have been introduced to improve transparency and trustworthiness. Clinical validation studies [121] highlight the gap between high experimental performance and practical deployment in primary care. As emphasized across systematic reviews [146], advancing this field will require larger, demographically balanced datasets, standardized preprocessing pipelines, and careful alignment between computational markers and clinically meaningful outcomes.

#### 3.8.5. Comparative Overview of Representative Studies

Table 8 provides an overview of the key characteristics of representative visual attention analysis studies.

## 4. Discussion

The rapid evolution of AI-driven methodologies for detecting ASD traits through voice and behavioral data underscores a paradigm shift in neurodevelopmental diagnostics. The present scoping review synthesizes advancements in the following eight modalities: voice biomarkers, linguistic and content-level language analysis, conversational/interactional dynamics, movement analysis, activity recognition, facial gesture analysis, visual attention and multimodal approaches., highlighting their strengths, limitations, and transformative potential. Notably, a significant proportion of the literature (25 studies) employs multimodal approaches, combining distinct data streams to enhance predictive power. This trend is exemplified by the seminal work of researchers, including Shamseddine et al. [148], Jaiswal et al. [111], Huynh et al. [107], Hassan et al. [149], Sellamuthu et al. [87], Wang et al. [150], Toki et al. [151], Rouzbahani et al. [152], Ouss et al. [153], Pouw et al. [154], Vabalas et al. [155], Almars et al. [114], Minissi et al. [133], Sapiro et al. [105], Wei et al. [156], Paolucci et al. [157], Deng et al. [158], Vidivelli et al. [159], Alam et al.(2020) [160], Alam et al. (2023) [72], Dedgaonkar et al. [94], Drimalla et al. [161], Drougkas et al. [162], and Natarajan et al. [163]. The capacity of these multimodal frameworks is particularly evidenced by Almadhor et al. (2025) [134], who achieved a notable integration of eye-tracking and kinematic data within a federated learning framework, utilizing deep neural networks (DNNs) and explainable AI (XAI) to attain model accuracies ranging from 88% to 99.99% across the datasets.


**Comparative Analysis of Modalities and the Path to Clinical Translation**


A critical synthesis of the literature, prioritizing the dual criteria of early detection potential and predictive accuracy, reveals a distinct hierarchy among the AI modalities investigated. The voice biomarker modality emerges as the most compelling avenue for future research. Its superiority is grounded in its unique capacity for extremely early signal acquisition, from infancy, coupled with a non-invasive nature and, as evidenced by this review, consistently high predictive performance. Following this, facial analysis and visual attention modalities also demonstrate significant promise, offering complementary behavioral data that align well with our core criteria.To transcend the limitations of any single modality and develop tools with the generalizability required for clinical deployment, a multimodal approach is paramount. We propose the following two strategic pathways: First, the integration of the highly promising voice biomarker with a visual modality, such as visual attention, to capture concurrent vocal and social engagement cues. Second, a powerful alternative involves fusing the closely related auditory-linguistic modalities, voice biomarkers, conversational dynamics, and language analysis, and combining this rich composite with one of the other modalities. Such integrative strategies are likely to yield the robust, generalizable, and clinically viable diagnostic tools necessary to improve outcomes through timely intervention.


**Key Advancements**


Diagnostic Accuracy: Machine learning (ML) and deep learning (DL) models consistently demonstrated high performance in controlled settings, with accuracies ranging from 85% to 99%. For instance, STFT-CNN models in voice analysis achieved 99.1% accuracy with augmented data, while gait analysis using 3DCNNs and MLPs classified ASD with 95% precision. These results suggest that AI can augment traditional tools like ADOS by offering objective, quantifiable metrics.Modality-Specific Insights:Voice Biomarkers: Acoustic features (pitch, jitter, and shimmer) and prosodic patterns emerged as robust indicators, though performance varied with linguistic and demographic diversity.Movement Analysis: Motor abnormalities (e.g., gait irregularities and repetitive movements) were effectively captured via wearable sensors and pose estimation tools, enabling low-cost, scalable screening.Facial Gesture Analysis: Vision Transformers and CNNs excelled in decoding atypical gaze patterns and micro-expressions, though dataset imbalances hindered subgroup differentiation.Multimodal Fusion: Integrating audio-visual, kinematic, and linguistic data (e.g., Deng et al. [30]’s LLaVA-ASD framework) improved robustness, addressing the heterogeneity of ASD presentations.Innovations in Explainability: Post hoc interpretability methods, such as attention maps in transformer models and feature importance scoring in RF/SVM, bridged the gap between “black-box” AI and clinical trust. For example, Briend et al. (2023) [10] linked shimmer and F1 frequencies to vocal instability in ASD, aligning computational outputs with clinical observations.


**Persistent Challenges**


Dataset Limitations: Small, homogeneous samples (e.g., gender imbalances and limited age ranges) and lack of multicultural/multilingual datasets restrict generalizability. For instance, Vacca et al. [2]’s voice analysis relied on 84 subjects, raising concerns about overfitting.Ethical and Practical Barriers: Privacy risks in video/audio data collection and algorithmic bias (e.g., underrepresentation of female and non-Western cohorts).Clinical Translation: Most studies operated in controlled environments, with limited validation in real-world clinical workflows. For example, Li et al. [82]’s MMASD dataset, while promising, lacks benchmarking against gold-standard diagnostic tools.


**Navigating the Ethical Landscape: From Bias to Actionable Guidance**


Our review identifies several critical ethical and practical barriers that must be proactively addressed to translate AI-based autism detection from research into responsible clinical practice. While the technical performance of these tools is promising, their real-world viability is contingent on overcoming these persistent challenges. Having only acknowledged these issues is insufficient. Therefore, we move beyond a general call for “ethical oversight” to propose the following concrete, actionable steps for the research community.

Mandatory Bias Auditing and Transparent Reporting: Future studies must mandate and report results of bias audits across diverse demographic subgroups. Model performance metrics (accuracy, sensitivity, etc.) should be disaggregated by sex, ethnicity, and socioeconomic background as a standard requirement for publication, moving beyond aggregate performance to ensure equitable efficacy.Development of Standardized Data Handling Protocols: To mitigate privacy risks, the field should develop and adopt standardized deidentification protocols for behavioral data. This includes technical standards for removing personally identifiable information from audio and video files, as well as clear guidelines for secure data storage, access, and sharing, similar to those used in genomic research.Promoting Explainability and Co-Design for Trust: To address the “black box” problem, a dual approach is essential. First, the development of explainable AI (XAI) techniques must be prioritized to provide clinicians with interpretable rationales for a model’s output (e.g., highlighting which specific vocal patterns or behaviors drove the assessment). Second, the development process must transition to a participatory design model, actively involving autistic individuals, their families, and clinicians. This collaboration is crucial for validating model explanations, ensuring they align with human experience, and building the foundational trust required for clinical integration.


**Future Directions**


Scalability Through Multimodal Integration: Combining voice, movement, and gaze data could capture the dynamic interplay of ASD traits, as seen in hybrid frameworks like 2sGCN-AxLSTM. Federated learning (Shamseddine et al., 2022 [148]) offers a pathway to pooling diverse datasets while preserving privacy.Longitudinal and Real-World Studies: Tracking developmental trajectories via wearable sensors or mobile apps (e.g., Wall et al. [164]’s gaze-tracking glasses) could enhance early detection and personalize interventions.Ethical AI Frameworks: Developing standardized benchmarks, inclusive datasets, and guidelines for consent/transparency is critical. Innovations like privacy-preserving optical flow (Li et al., 2023 [65])and synthetic data generation may mitigate biases.

## 5. Conclusions

This scoping review highlights AI’s transformative potential in augmenting conventional ASD diagnostic frameworks, particularly through voice- and behavior-based modalities. Voice biomarkers and facial gesture analysis have demonstrated particular promise, with models achieving near-perfect accuracy in ideal conditions, making ASD detection more accessible, faster, and potentially available in resource-limited settings. However, to fully realize this potential, future research must address several critical needs, outlined as follows:Expansion of datasets that are multilingual, diverse in age/gender, and inclusive of varying ASD severities.Development of standardized pipelines and benchmarks for feature extraction, model training, and evaluation.Stronger emphasis on model interpretability and clinical usability.Rigorous ethical oversight and privacy protection, particularly in the use of video and audio recordings.

Additionally, collaborative efforts between clinicians, AI researchers, and policymakers are essential to bridge the gap between experimental success and clinical deployment. Longitudinal studies and real-world validations will be key to building AI systems that are not only accurate but also trusted, fair, and generalizable across populations.

## Figures and Tables

**Figure 1 bioengineering-12-01136-f001:**
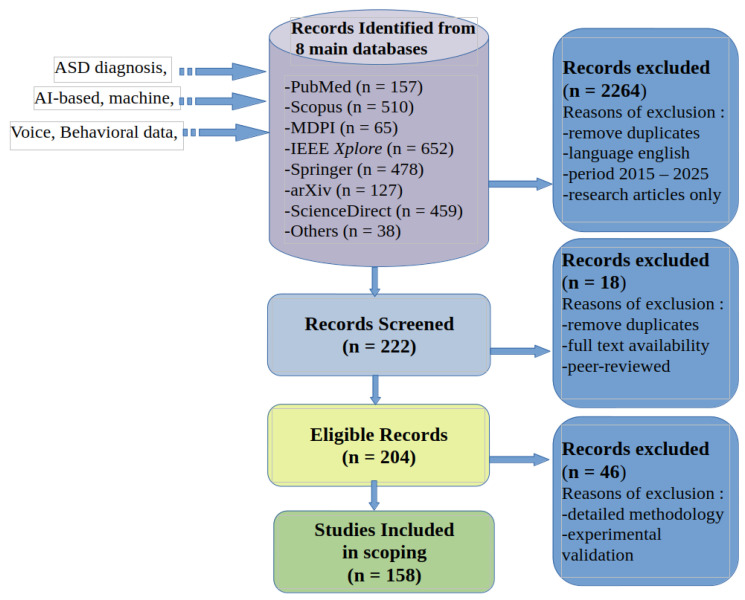
PRISMA-ScR flow diagram for the article selection process.

**Figure 2 bioengineering-12-01136-f002:**
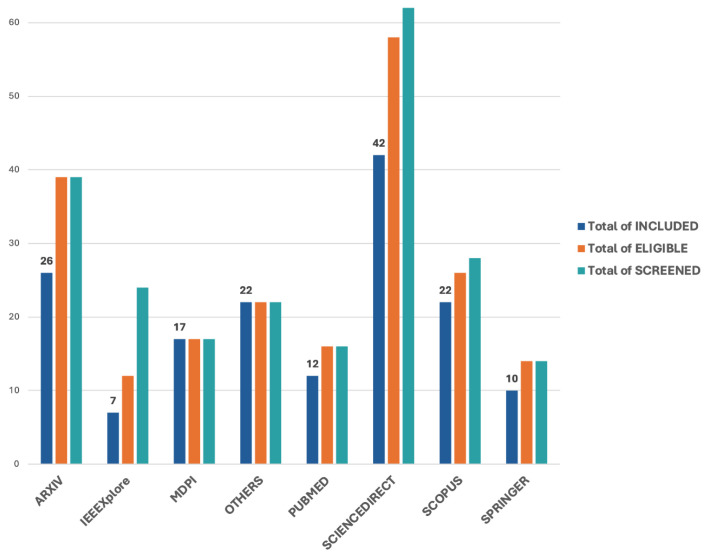
Total of research articles per database: included, eligible, and screened.

**Figure 3 bioengineering-12-01136-f003:**
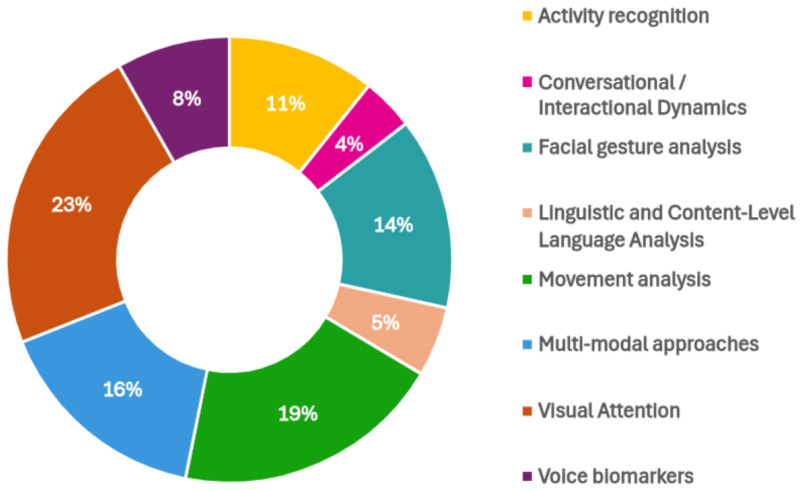
Percentage of research articles included by modality.

**Figure 4 bioengineering-12-01136-f004:**
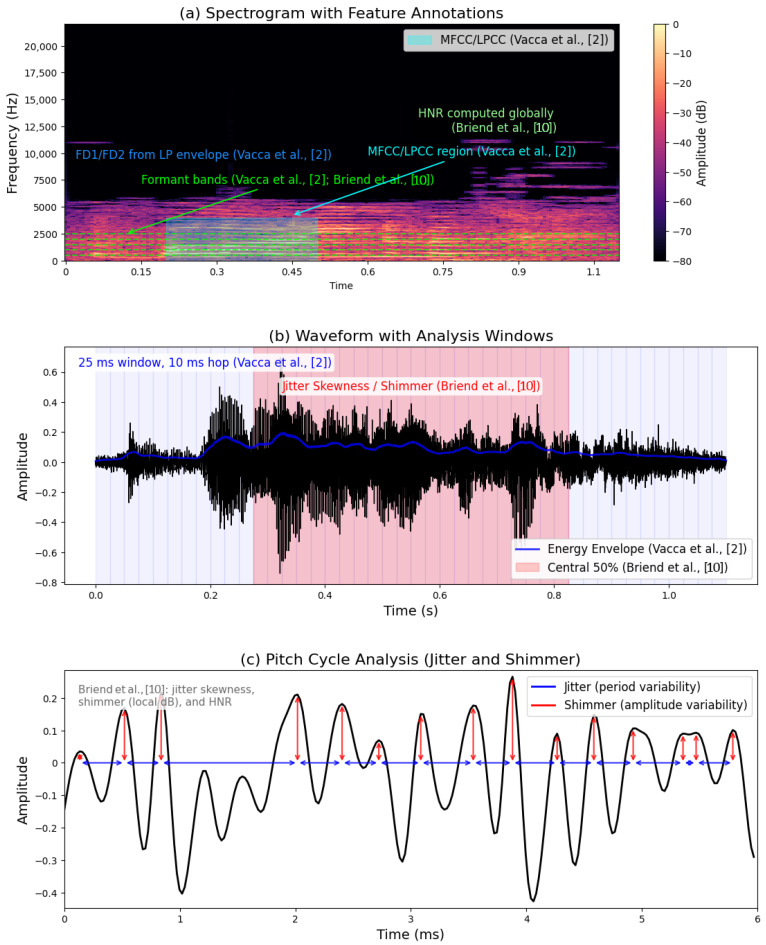
Technical comparison of speech signal features: Vacca et al. [2] vs. Briend et al. [10]. (**a**) Spectrogram with feature regions, (**b**) windowed feature extraction, and (**c**) cycle-level jitter and shimmer.

**Figure 5 bioengineering-12-01136-f005:**
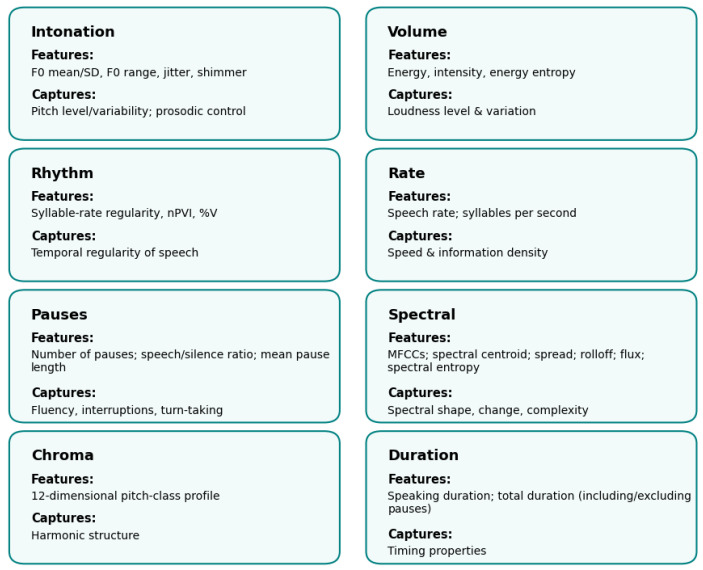
Hu et al. [11], categorization of speech features in clinical autism assessments.

**Table 1 bioengineering-12-01136-t001:** Research articles included with their performance by modality.

Modality	Articles Included	Common Algorithms	Avg. Accuracy/ Performance	Sample Size of the Best Accuracy
Voice Biomarkers	13	SVM, RF, k-Means	85–99%	30 children
Linguistic and Content-Level Language Analysis	8	BERT, LSTM, Transformer-based NLP XGBoost	80–97%	120 children
Conversational/ Interactional Dynamics	6	HMM, Dialogue models, Graph-based models	76-90%	79 children
Movement Analysis	31	3DCNN, LSTM, RQA	90–95%	118 children
Activity Recognition ^1^	17	SVM, RF, k-Means	87–93%	120 children
Facial Gesture Analysis ^2^	22	Vision Transformers, CNNs	83–96%	2926 facial images
Visual Attention	36	Eye-tracking, Attention models, CNN-RNN	88–97%	547 facial images (59 children)
Multimodal Approaches	25	Fusion models, Ensemble deep learning, Transformer hybrids	70–99%	44 samples of kinematic and eye movement features

^1^ Restricted and Repetitive Behaviors (RRBs) and behavior in response to a stimulus. ^2^ Eye gaze not included but part of the Visual Attention modality.

**Table 2 bioengineering-12-01136-t002:** Comparison of studies on voice biomarkers for ASD detection (acoustic/prosodic features only).

Authors	Dataset Size	Age	Data Preprocessing	Feature Extraction	AI Model/Tool
Mohanta & Mittal (2022) [13]	33 children	3–9	Cleaning, segmentation, normalization	Formants, MFCCs, jitter	GMM, SVM, RF, KNN, PNN (98%)
Briend et al. (2023) [10]	84 children	8–9	Voice segmentation	Prosodic, MFCCs, pitch, intensity	SVM, RF, XGBoost (90%)
Vacca et al. (2024) [2]	84 children	6–12	Noise reduction, normalization	Acoustic (pitch, formants)	SVM, RF (98.8%)
Jayasree & Shia (2021) [14]	19 children	5–14	Denoising, feature scaling	MFCCs, jitter, shimmer	FFNN (87%)
Asgari & Chen (2021) [15]	118 children	7–15	Cleaning, segmentation	Pitch, jitter, shimmer, MFCCs, prosody	SVM, RF (86%)
Godel et al. (2023) [16]	74 children	2–6	Segmentation, normalization, annotation	Intonation, rhythm, stress	ML classifiers (85%)
Keerthana Sai et al. (2024) [12]	30 children	3.5–9	STFT, normalization	STFT spectrograms, pitch	Custom CNN (99.1%)
Chi et al. (2023) [22]	58 children	3–12	Cleaning, segmentation	MFCC, spectral, raw audio	RF, wav2vec 2.0, CNN (79%)
Guo et al. (2022) [17]	40 children	4–9	Cleaning, tone segmentation	Jitter, shimmer, HNR, MFCCs	RF (78%)
Hu et al. (2024) [11]	44 children	4–16	Cleaning, segmentation, normalization	Prosody, pitch, duration, rhythm	SVM, RF, CNN (89%)

**Notes:** ASD = Autism Spectrum Disorder; MFCC = Mel-Frequency Cepstral Coefficients; HNR = Harmonic-to-Noise Ratio; XGBoost = eXtreme Gradient Boosting. Tool abbreviations: RF = Random Forest; SVM = Support Vector Machine; PNN = Probabilistic Neural Network; FFNN = Feed-Forward Neural Network.

**Table 3 bioengineering-12-01136-t003:** Representative studies on conversational/interactional dynamics in ASD.

Authors	Dataset Size	Age	Data Preprocessing	Feature Extraction	AI Model/Tool
Ochi et al. (2019) [23]	79 adults (HF-ASD, TD)	19–36	Audio segmentation; speaker diarization; pause detection	Turn-gap duration; pause ratio; blockwise intensity synchrony	Statistical modeling; SVM 89.9%
Lehnert-LeHouillier et al. (2020) [24]	24 children/ teens	9–15	Segmentation of natural conversation; prosody extraction (Praat)	Prosodic entrainment (F0, intensity, rate)	Linear regression models
Lau et al. (2022) [18]	146 children- adults	6–35	Phonetic alignment, segmentation	Intonation, stress, duration	SVM (83%)
Wehrle et al. (2023) [25]	40 adults	18–45	Free-dialogue segmentation; pause detection	Turn-timing distributions; gap preferences	Distributional analysis
Yang et al. (2023) [27]	72 children (therapy sessions)	3–11 years	Therapist–child session segmentation; diarization	Turn/response timing; prosodic cues for conversation-quality prediction	Supervised ML & DL
Chowdhury et al. (2024) [26]	29 children (ADOS-style)	10-15 years	Examiner–child dialogue segmentation; prosody + lexical tags	Turn counts; latency; basic prosody	RF (acc. 76%)

**Table 4 bioengineering-12-01136-t004:** Representative studies on language and content-level analysis for ASD detection.

Authors	Dataset Size	Age	Preprocessing	Feature Extraction	AI Model/Tool
Mukherjee et al. (2023a) [28]	1200 text samples	5–16	Cleaning, tokenization	BERT, GPT-3.5 embeddings	BERT, GPT-3.5, SVM (92%)
Rubio-Martín et al. (2024) [29]	2500 text samples	6–18	Normalization, lemmatization	TF-IDF, embeddings, sentiment	SVM, CNN, LSTM (90%)
Murugaiyan & Uyyala (2023) [33]	2000 customer speech (text)	—	Speech-to-text, cleaning	Sentiment, embeddings	Deep CNN + BiLSTM (90%)
Deng et al. (2024) [30]	800 children	4–14	Speech-to-text, alignment	Semantics, syntax, behavior cues	Hybrid DL (89%)
Angelopoulou et al. (2024) [31]	600 narrative samples	5–12	Transcription, cleaning	Lexical diversity, cohesion	RF, SVM (89%)
Ramesh et al. (2023) [34]	86 children	1–6	Cleaning, encoding, SMOTE	Mean Length of Utterance and Turn (MLU, MLT ratio), POS	LR, SVM, KNN, NB, RF (80%)
Jaiswal et al. (2024) [35]	17,323 ASD and 171,273 users	–	Cleaning, tokenization, stemming, lemmatization, encryption	Bag-of-words Term Frequency (TF-IDF), Word2vec	Logistic regression, Bi-LSTM (F1 80%)
Themistocleous et al. (2024) [5]	120 children	4–10	Transcription, Random over-sampling, cleaning	Grammatical, semantic, syntactic and text complexity features	Gradient Boosting, Decision trees, Hist GB, XGBoost (97.5%)

**Table 5 bioengineering-12-01136-t005:** Representative movement analysis studies for ASD (includes video, depth, motion capture, and wearable-sensor works focused on motor/postural/gait assessment).

Authors	Dataset Size	Age	Data Preprocessing	Feature Extraction	AI Model/Tool
Al-Jubouri et al. (2020) [66]	Kinect 3D gait clips for 118 children	4–12 years	Gait-cycle isolation; MICE imputation; face blurring; augmentation	3D kinematics; PCA reduction	MLP (acc. 95%)
Li et al. (2023) (MMASD) [65]	Therapy videos (>100 h) for 32 children	5–12	Clip segmentation; noisy clip removal; privacy streams	Optical flow; 2D/3D skeletons; clinical scores	Benchmark dataset
Zhang et al. (2021) [67]	ASD action videos	5–10 years	Pose extraction; sequence curation	Skeleton sequences; temporal features	LSTM (acc. 93.55%
Serna-Aguilera et al. (2024) [39]	Video responses to stimuli of 66 subjects	Children	Frame/clip selection; normalization	Frame/flow deep features	CNN (acc. 81.48%
Sadek et al. (2023) (Head-banging) [68]	SSBD + collected videos	4–12	Pose estimation; clip curation	Skeleton + movement patterns	NN/CNN (acc. 85.5% to 93%)
Sadek et al. (2023) (Arm-flapping) [69]	Recorded videos	Children	Video preprocessing; pose estimation	Skeleton-based stereotypy cues	CNN/LSTM
Li et al. (2020) (Postural control) [54]	Lab postural videos for 50 children	5–12 years	Trial segmentation; normalization	Postural sway/kinematics	6 MLs, Naïve Bayes best acc. 90%
Wei et al. (2023) [71]	SSBD dataset- 61 videos	61 subjects	Activity segmentation; normalization	Vision-based activity cues	RGB I3D + MS-TCN (acc. 83%)
Ganai et al. (2025) [63]	Multi-site gait data-61 children	4–6 mean years	Gait trial segmentation; QC	Spatiotemporal gait deviations	SVM/RF/Logit best acc. 82%
Zhao et al. (2022) (Head movement) [64]	Clinical videos of 43 children	6–13 years	Head-track preprocessing	Head-movement features	ML (DT/SVM/RF acc. 92.11%
Doi et al. (2022) (Infant) [47]	62 mother-infant videos	4–18 months	Markerless capture; QC	Spontaneous movement metrics	LDA, MLP
Bruschetta et al. (2025) (Infant) [43]	74 infants-Home/clinic videos	10 days to 24 weeks	Standardized capture; normalization	Markerless infant kinematics	SVM 85%

**Notes:** MICE = Multiple Imputation by Chained Equations; SSBD = Self-Stimulatory Behavior Dataset; QC = quality control; PCA = principal component analysis; CNN = convolutional neural network; LSTM = long short-term memory; MLP = multilayer perceptron; RF = random forest; SVM = support vector machine; ML = machine learning; and LDA = linear discriminant analysis.

**Table 6 bioengineering-12-01136-t006:** Comparison of representative autism activity recognition classification studies.

Authors	Dataset Size	Age	Data Preprocessing	Feature Extraction	AI Model/Tool
Zhao et al. (2025) [81]	120 children (ASD/control), AV–FOS audio–video	3–12	A/V sync; segmentation; normalization	Multimodal interaction cues (prosody; gesture; facial; engagement)	Transformer-based AV model (acc. 93%)
Singh et al. (2024) [73]	80 children (ASD/control), gesture videos	4–12	Video augmentation; YOLOv7 hand detection; frame normalization	Gesture pose/motion; temporal patterns	YOLOv7 + VideoMAE (acc. 92%)
Deng et al. (2024) [30]	200 children (ASD/control), A/V recordings	3–12	A/V sync; denoising; segmentation; normalization	Audio: prosody, MFCC; Visual: pose, gesture, facial	Multimodal DL (CNN video, RNN audio, fusion) (acc. 91%)
Liu & He (2021) [83]	3000+ samples, multimodal	2–16	Data fusion; normalization; feature selection	Behavioral; imaging; genetic; clinical	Ensemble (SVM; RF; DNN) (acc. 91%)
Asmetha & Senthilkumar (2025) [75]	80 children, stereotypy videos	4–14	Video segmentation; augmentation; normalization	Spatiotemporal stereotypy features	Dual self-supervised Video Transformer (acc. 91%)
Alam et al. (2023) [72]	800 meltdown episodes	5–16	A/V segmentation; normalization; augmentation	Audio; video; physiological meltdown cues	CNN–LSTM hybrid (acc. 90%)
Li et al. (2024) [82]	120 dyads, block-play protocol (video)	2–6	Segmentation; pose estimation; normalization	Dyadic interaction features	2sGCN–AxLSTM hybrid (acc. 89%)
Jayaprakash & Kanimozhiselvi (2024) [77]	182 clinical/ behavioral records	2–12	Cleaning; normalization; encoding	Clinical/ behavioral items	Multinomial logistic regression (acc. 88%)
Jin et al. (2023) [46]	80 children (ADOS videos)	3–8	Video segmentation; pose estimation; normalization	Activity level metrics	SVM; RF; Gradient Boosting (RF best acc. 87%)
Lakkapragada et al. (2022) [74]	50 children, hand-movement videos	4–10	Video augmentation; hand extraction; normalization	Abnormal hand movement metrics	RF; SVM (best acc. 87%)

**Table 7 bioengineering-12-01136-t007:** Top 10 accuracies: Comparison of autism facial gesture analysis classification studies.

Authors	Dataset Size	Age	Data Preprocessing	Feature Extraction	AI Model/Tool
Ibadi et al. (2025) [4]	2926 facial images	Children	Normalization, augmentation,	Static features by using Vision Transformer (VIT) model and Squeze-and-Excitation (SE) blocks	Xception, ResNet50, VGG-19, MobileNetV3, EfficientNet-B4, ASDvit model (AUC 97.7%)
Beno et al. (2025) [84]	2726 facial images	2–14 years	Normalization with magnitude stretching, augmentation,	Static facial features by using DenseNet and ResNet architectures	DenseResNet model (AUC 97.07%)
Ruan et al. (2024) [86]	42 ADOS-2 interview videos	16–37 years	Frame extraction, Face detection, Motion magnification	Micro-expression features by using SOFTNet and Micron-BERT model	MLP, SVM, ResNet (acc. 94.8%)
Almars et al. (2023) [114]	1100 facial images	—	Face detection, normalization, augmentation	Deep features via transfer learning	CNN with Gorilla Troops Optimizer (acc. 94.2%)
Shahzad et al. (2024) [85]	2000 facial images	3–16 years	Face detection, normalization, augmentation	Landmarks, emotion, attention maps	Hybrid attention deep model (acc. 94%)
Farhat et al. (2024) [113]	1200 facial images	3–15 years	Face detection, normalization, augmentation	Deep CNN embeddings	CNN, ResNet, VGG (best acc. 93%)
Gaddala et al. (2023) [92]	2000 facial images	4–16 years	Face detection, normalization, augmentation	Deep facial conv features	Deep CNN (acc. 93%)
Gautam et al. (2023) [109]	1000 facial images	3–14 years	Face detection, normalization, augmentation	Landmarks, geometric, texture	YOLOv8 vs. CNN/SVM (YOLOv8 best acc. 93%)
Akter et al. (2021) [99]	2000 facial images	3–14 years	Face detection, normalization	Deep facial embeddings	Transfer learning (VGG, ResNet, best acc. 93%)
ElMouatasim & Ikermane (2023) [91]	1200 facial images	3–15 years	Face cropping, normalization	Deep CNN embeddings	ResNet, VGG (best acc. 92%)

**Table 8 bioengineering-12-01136-t008:** Representative visual attention analysis studies for ASD (screen-based, dyadic/wearable, VR, and multimodal gaze-focused works).

Authors	Dataset Size	Age	Data Preprocessing	Feature Extraction	AI Model/Tool
Pierce et al. (2023) [120]	N≈653 toddlers	12–48 months	ROI segmentation; task standardization; calibration QC	% gaze to motherese vs. geometric patterns	Threshold/ML rule
Chang et al. (2021) [119]	N=993 (ASD = 40, TD = 936, DDLD = 17)	16–30 months	Mobile eye-tracking calibration; synchronization	Social vs. nonsocial fixation; speech–gaze coordination	ML classifiers (AUC ≈ 0.88–0.90)
Fang et al. (2021) [118]	Saliency4ASD (300 images for training, 200 for benchmark)	5–12 years	Images expanded via rotation and flipping; Pre-training on other datasets and fine-tuning	Spatial Feature Module (SFM) with FCN; Pseudo Sequential Feature Module (PSFM) with two ConvLSTMs	Two-stream model (ASD-HSF)
Xie et al. (2022) [3]	N=39 (20 ASD, 19 TD)	mean age ∼31 years	Fixation maps; Gaussian smoothing; normalization; augmentation	Two-stream integration of images and fixation maps; discriminative feature analysis	VGG-16 two-stream CNN (ASDNet); acc. up to 0.95 (LOO CV)
Chong et al. (2020) [123]	N=103 (ASD = 66, TD = 55; 18 to 60 mo; plus 15 ASD, 5–13 y)	18 months– 13 years	Egocentric video capture; stabilization; alignment	Eye-contact episode detection	CNN (ResNet-50; wearable PoV)
Zhao et al. (2021) [124]	N=39 (ASD = 19, TD = 20)	Eye-tracking (face-to-face conversation)	Fixation time on AOIs (eyes, mouth, face, body); session length	Visual attention during structured conversation	SVM, LDA, DT, RF (SVM best, 92.3%)
Alcañiz et al. (2022) [122]	N=55 (ASD = 35, TD = 20)	4–7 years	VR eye-tracker calibration; AOI feature extraction	VR-based fixation metrics (social vs. nonsocial; faces vs. bodies)	ML classifiers (SVM, RF, kNN, NB, XGBoost)
Minissi et al. (2024) [133]	N=81 (ASD = 39, TD = 42)	3–7 years	VR calibration; biosignal recording (CAVE)	Motor skills, eye movements, behavioral responses	Linear SVM (RFE, nested CV)
Wei et al. (2024) [125]	N=529 (ASD = 290, TD = 239)	1.5–6 years	Eye-tracking paradigms (social + non-social tasks; calibration; Z-score normalization)	Fixation, saccade, pursuit, joint attention features	RF (best), SVM/LR/ANN/ XGB (comparisons)
Cilia et al. (2021) [126]	N=59 (ASD = 29, TD = 30)	mean age ∼7–8 years	Scanpath visualization; data augmentation	Eye-tracking scanpath images with dynamics	CNN (4 conv + pooling, 2 FC)
Ahmed et al. (2022) [128]	N=59 (ASD = 29, TD = 30)	Children	Image enhancement (average + Laplacian filters); ROI segmentation	LBP + GLCM features; deep feature maps	ANN/FFNN; GoogLeNet; ResNet-18; CNN+SVM hybrids
Liaqat et al. (2021) [127]	N=28 (ASD = 14, TD = 14)	6–12 years	Scanpath normalization; statistical features	Synthetic saccades + fixation maps	MLP/BrMLP, CNN, LSTM
Keehn et al. (2024) [121]	N=154 (146 usable)	ch. and Todd.	AOI-based calibration QC; blink removal; interpolation; artifact filtering	Multiple eye-tracking biomarkers (GeoPref, PLR, gap-overlap, fixation metrics)	Composite biomarker + logistic regression
Antolí et al. (2025) [129]	N=93 (ASD = 24, DLD = 25, TD = 44)	32–74 months	AOI-based preprocessing; eye-tracking metrics extraction	Social vs. non-social fixation features	Naive Bayes, LMT (XAI)
Jiang & Zhao (2017) [130]	N=39 (20 ASD, 19 TD)	Adults	Eye-tracking with natural-scene free viewing; Fisher-score image selection	DoF maps + DNN features (VGG-16)	DNN + SVM
Almadhor et al. (2025) [134]	N>100 multimodal	Children	Multimodal calibration; sync	Gaze + kinematic features	Explainable ML (XAI)
Tao & Shyu (2019) [131]	Saliency4ASD (300 images; 14 ASD + 14 TD)	Children	Saliency map patch extraction; duration integration	Scanpath-based saliency patches + durations	CNN–LSTM
Mazumdar et al. 2021) [132]	Saliency4ASD (28 children)	Children	Object/non-object detection; fixation	Visual behavior features	Ensemble (TreeBagger)
Vasant et al. (2024) [115]	N=53 (ASD = 26, TD = 27)	3–6 years	Tablet-based calibration; preprocessing of gaze trajectories	Saliency maps + trajectory-based features	Transfer learning CNN models
Alsaidi et al. (2024) [135]	Public dataset (59 children; 547 images)	Children (∼8 years)	Image resizing; grayscale conversion; dimensionality reduction	Eye-tracking scan path images	CNN (T-CNN-ASD)
Sá et al. (2024) [137]	N=86	3–18 y (Ch./Adol.)	Joint-attention paradigm; floating ROI + resampling	Anticipation + gaze fixation features	Heterogeneous stacking ensemble (F1 = 95.5%)
Ozdemir et al. (2022) [138]	N=133 (ASD = 61, TD = 72)	26–36 mo	Eye-tracking preprocessing; feature selection (ReliefF, IG, Wrapper)	Fixation/dwell/ diversion duration features (social vs. non-social AOIs)	ML classifiers (DT, NB, RF, SVM) for DSS
Varma et al. (2022) [139]	N=95 (ASD = 68, NT = 27)	2–15 y	Automated gaze annotation; AOI discretization	Gaze fixation + visual scanning patterns	LSTM (mild predictive power)
Thanarajan et al. (2023) [140]	N=547 (ASD = 219, TD = 328)	Ch.	U-Net segmentation; CBO-based hyperparameter tuning	Inception v3 features + LSTM	DL framework (ETASD-CBODL)
Islam et al. (2025) [141]	Public datasets (TD = 628 images, ASD = 519 images)	Ch.	Data augmentation; fixation map preprocessing	Hybrid involution–convolution gaze features	Lightweight deep model (3 Involution + 3 Convolution layers)
Solovyova et al. (2020) [142]	N≈100 (ASD = 50, TD = 50), 3–10 y	Ch.	Calibration; Gaussian noise simulation	Time distribution across face ROIs (eyes, mouth, periphery)	Fully connected NN classifier
Fernández et al. (2020) [143]	N=31 (ASD = 8, TD = 23)	2–6 y	Face/eye detection; gaze labelling; augmentation	Gaze direction (social vs. abstract vs. undetermined)	CNN (LeNet-5)
Shi et al. (2025) [144]	Qiao (74: 50 ASD, 24 TD, 5 y), Cilia (59: 29 ASD, 30 TD, 3–12 y), Saliency4ASD (28: 14 ASD, 14 TD, 5–12 y)	Ch.	Invalid gaze removal; clustering with KMeans, KMedoids, AC, BIRCH, DBSCAN, OPTICS, GMM	Internal cluster validity indices (SC, CH, DB, newDB, DI, CSL, GD33, PBM, STR)	ML models (LR, SVM, KNN, Decision Tree, RF, XGBoost, MLP) for ASD prediction (AUC up to 0.834)
Yu et al. (2024) [145]	N=52 (ASD = 43, TD = 9)	Ad. (age 18–28 y)	Video preprocessing; speech–visual sync	Gaze engagement, variance, density, diversion	Random Forest (video-based features)
Colonnese et al. (2024) [147]	N=28, 5–12 y	Ch.	Decomposition; augmentation	TracIn (outlier removal + influential samples)	CNN + LSTM (GBAC)
Kanhirakadavath & Chandran (2022) [136]	N=59 (30 ASD, 29 TD; 547 images; 3–13 y)	3–13 y	ETSP image preprocessing; PCA/CNN feature extraction	Eye-tracking scan path images	BDT, DSVM, DJ, DNN (CNN+DNN best, AUC = 97%)

**Notes:** ROI = region of interest; CNN = convolutional neural network; LSTM = long short-term memory; RF = random forest; SVM = support vector machine; QC = quality control; ML = machine learning; XAI = explainable AI. Ages: mo = months; y = years. Ch. = children; Adol. = adolescents; and Ad. = adults.

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
