# Peer review of "A Scoping Review of AI-Based Approaches for Detecting Autism Traits Using Voice and Behavioral Data"

_bioengineering, 2025, doi:10.3390/bioengineering12111136_

Round 1

Reviewer 1 Report

Comments and Suggestions for Authors

The review "A Scoping Review of AI-Based Approaches for Detecting Autism Traits Using Voice and Behavioral Data" have potential to be published. The highlights of the article are that it is a very informative text and organized into several tables that can help the reader in their search for information. Furthermore, the review shows a methodology well described with current references (last 10 years) and whose selection was well justified. However, I think that in beggining, the text is very concerned with detailing the number of articles that has been published for each of the research modalities and focusing less on other matters. I believe the review would be even more complete with a section explaining in more detail the ASD discrimination criteria for each of these eight modalities from a clinical point of view.

Observations and suggestions:

- Reduce the Abstract to 200 words. I counted 252 words. I suggest exclude abbreviations: "(MRI, fMRI, DTI, EEG)" (line 3) and "(e.g., ADOS, M-CHAT, ADI-R)" (line 4), because there are rules about the use of abbreviations in MDPI guidelines (https://www.mdpi.com/authors/layout#_bookmark5)  

- In the paragraph "Recent advances in artificial intelligence (AI) have facilitated the development of innovative tools for the early detection of autism traits that analyze various data modalities, including voice characteristics, MRI/fMRI and brain imaging, EEG signals, eye-tracking, facial expressions, and both verbal and non-verbal cues." (lines 43-46), I would include some references already in this paragraph. Although the review cites several references throughout the text, it is interesting include the citation of some references already in this paragraph. This would help the reader in the search for references on the use of AI in the development of ASD detection methods for each of these modalities, right at the beginning of the review.

- The Figure 4 need be introduced in the text. The Figure 4 suddenly appeared in the manuscript. Furthermore, it is necessary a more detailed explanation of spectrograms and how they are used and interpreted as a method for ASD detection. 

- I suggest add some few references with examples of the use of AI to detect others mental disorders (such as schizophrenia, for example) through behavioural dataset. This helps to demonstrate the contribution of AI to the development of general clinical methods.   

Minor corrections:

General observation: Remember to explain the abbreviation the first time it appears in the main text, after that use only the abbreviation.

- In Introduction (line 29) change "ASD is a complex..." to "Autism spectrum disorder (ASD) is a complex...", because that the abbreviation need be the first time explained in the main text;  
- Change "social interaction" to "social interactions" (line 30);
- Change "EEG signals" to "electroencephalography (EEG) signals" (line 45);
- Change "Voice Bio-Markers" to "Voice Biomarkers" (subtitle 3.2, line 135) to be standardized with all the text;
- Change "“voice biomarkers” to "“voice bio-markers” (line 138), since you are now talking about the other terms used in the research, but note that in the text of your manuscript you used biomarkers (without hyphen) as the main term;
- Change "MFCCs" to "Mel-Frequency Cepstral Coefficients (MFCCs)" (line 158);
- Change "Vacca et al.[1] tested RF, SVM,..." to "Vacca et al.[1] tested random forests (RF), support vector machine (SVM),..." (line 200). In the same way, "support vector machines" can be deleted from lines 305 and 492 and used only "SVM" and "random forests" can be deleted from lines 306 and 492 and used only "RF";
- Change "ADHD" to "Attention-Deficit / Hyperactivity Disorder" (line 226);
- Change "LENA-based pipelines" to "Language ENvironment Analysis (LENA)-based pipelines" (line 275). Exclude "(Language ENvironment Analysis)" in line 332;
- Change "TF-IDF" to "Term Frequency-Inverse Document Frequency (TF-IDF)" (line 351)
- Change "LSTM" to "long short-term memory (LSTM)" (line 363) and use only LSTM in line 499. 

Author Response

Comments 1: The review "A Scoping Review of AI-Based Approaches for Detecting Autism Traits Using Voice and Behavioral Data" have potential to be published. The highlights of the article are that it is a very informative text and organized into several tables that can help the reader in their search for information. Furthermore, the review shows a methodology well described with current references (last 10 years) and whose selection was well justified. However, I think that in beggining, the text is very concerned with detailing the number of articles that has been published for each of the research modalities and focusing less on other matters. I believe the review would be even more complete with a section explaining in more detail the ASD discrimination criteria for each of these eight modalities from a clinical point of view.

Response 1: Thank you for pointing this out. We agree with this comment. Therefore, we have created Section 1.3. ASD discrimination criteria (page …) to highlight the ASD discrimination criteria for each of these eight modalities from a clinical point of view.

Comments 2: Reduce the Abstract to 200 words. I counted 252 words. I suggest exclude abbreviations: "(MRI, fMRI, DTI, EEG)" (line 3) and "(e.g., ADOS, M-CHAT, ADI-R)" (line 4), because there are rules about the use of abbreviations in MDPI guidelines (https://www.mdpi.com/authors/layout#_bookmark5)

Response 2: Agree. We have, accordingly, synthesized the abstract to fit in 200 words.

Comments 3: - In the paragraph "Recent advances in artificial intelligence (AI) have facilitated the development of innovative tools for the early detection of autism traits that analyze various data modalities, including voice characteristics, MRI/fMRI and brain imaging, EEG signals, eye-tracking, facial expressions, and both verbal and non-verbal cues." (lines 43-46), I would include some references already in this paragraph. Although the review cites several references throughout the text, it is interesting to include the citations of some references already in this paragraph. This would help the reader in the search for references on the use of AI in the development of ASD detection methods for each of these modalities, right at the beginning of the review.

Response 3: We agree with these comments and included four references in the paragraph at lines 40 to 42.

Comments 4: The Figure 4 need be introduced in the text. The Figure 4 suddenly appeared in the manuscript. Furthermore, it is necessary a more detailed explanation of spectrograms and how they are used and interpreted as a method for ASD detection.

Response 4:  We thank the reviewer for this valuable feedback. We have now taken the following actions to address the points raised: Figure 4 (a) and (c) are introduced in the text at line 274; and Figure 4 (b) is introduced at line 275.

Comments 5: - I suggest add some few references with examples of the use of AI to detect others mental disorders (such as schizophrenia, for example) through behavioural dataset. This helps to demonstrate the contribution of AI to the development of general clinical methods.

Response 5: We agree with your point of view, so we include these sentences with some references at line 42 to 46: “Similar AI-based approaches have also been explored to support the understanding and detection of other neurodevelopmental conditions through behavioural dataset, such as Attention-Deficit Hyperactivity Disorder (ADHD) [164 ] and Cerebral Palsy [166 ], as well as to assist in the assessment of psychiatric disorders [165 ], including schizophrenia, anxiety disorders, depression, and bipolar disorder.” 

[165] :  Schizophrenia: A Survey of Artificial Intelligence Techniques Applied to Detection and Classification https://www.mdpi.com/1660-4601/18/11/6099

Comments 6: Minor corrections

General observation: Remember to explain the abbreviation the first time it appears in the main text, after that use only the abbreviation.

- In Introduction (line 29) change "ASD is a complex..." to "Autism spectrum disorder (ASD) is a complex...", because that the abbreviation need be the first time explained in the main text;  

- Change "social interaction" to "social interactions" (line 30);

- Change "EEG signals" to "electroencephalography (EEG) signals" (line 45);

- Change "Voice Bio-Markers" to "Voice Biomarkers" (subtitle 3.2, line 135) to be standardized with all the text;

- Change "“voice biomarkers” to "“voice bio-markers” (line 138), since you are now talking about the other terms used in the research, but note that in the text of your manuscript you used biomarkers (without hyphen) as the main term;

- Change "MFCCs" to "Mel-Frequency Cepstral Coefficients (MFCCs)" (line 158);

- Change "Vacca et al.[1] tested RF, SVM,..." to "Vacca et al.[1] tested random forests (RF), support vector machine (SVM),..." (line 200). In the same way, "support vector machines" can be deleted from lines 305 and 492 and used only "SVM" and "random forests" can be deleted from lines 306 and 492 and used only "RF";

- Change "ADHD" to "Attention-Deficit / Hyperactivity Disorder" (line 226);

- Change "LENA-based pipelines" to "Language ENvironment Analysis (LENA)-based pipelines" (line 275). Exclude "(Language ENvironment Analysis)" in line 332;

- Change "TF-IDF" to "Term Frequency-Inverse Document Frequency (TF-IDF)" (line 351)

- Change "LSTM" to "long short-term memory (LSTM)" (line 363) and use only LSTM in line 499.

Response 6: We agree with these 11 points. And corrections have been made at lines: 24, 25, 40, 237 (subtitle 3.2), 240, 260, 303, 329, 379, 453, 466.

Reviewer 2 Report

Comments and Suggestions for Authors

This paper presents a comprehensive scoping review of AI-based methods for detecting autism traits using voice and behavioral data. It systematically analyzes 158 studies published from 2015 to 2025, summarizing advances across eight behavioral modalities and identifying key methodological challenges such as dataset bias and small sample sizes. The study provides a valuable overview of current progress and outlines directions for developing more inclusive and clinically applicable diagnostic tools.
Decision: Minor Revision.

While the paper offers a broad and informative synthesis of existing studies, the methodology for identifying and categorizing the 158 reviewed papers could be described in greater detail. The authors should clarify the inclusion and exclusion criteria, database sources, and the specific process used to group studies into the eight behavioral modalities. Providing a transparent description of the review protocol (e.g., PRISMA flow diagram or equivalent) would enhance the reproducibility and credibility of the review.

Although the review summarizes the overall trends and limitations of AI-based ASD detection studies, the discussion section would benefit from a deeper comparative analysis. Specifically, the authors could critically evaluate which modalities (e.g., voice vs. movement vs. multimodal data) have shown the most clinical potential and why. Highlighting cross-modal integration challenges and suggesting how future work could bridge gaps between technical performance and clinical applicability would strengthen the contribution of the paper.

Author Response

Comments 1: While the paper offers a broad and informative synthesis of existing studies, the methodology for identifying and categorizing the 158 reviewed papers could be described in greater detail. The authors should clarify the inclusion and exclusion criteria, database sources

Response 1: We thank the reviewer for this valuable feedback. We agree that a more detailed description of the methodology strengthens the review's rigor and reproducibility. In response, we have thoroughly revised the 'Materials and Methods' section to provide a comprehensive, step-by-step account of our process from lines 162 to 215.

Specifically, we have:

1/ Explicitly detailed the inclusion and exclusion criteria in the newly created subsection '2.3. Eligibility'. This subsection now clearly lists the specific requirements a study had to meet to be included (e.g., peer-reviewed, full-text available, primary focus on AI for ASD detection) and the reasons for exclusion.

2/ Explicitly listed all database sources and search platforms used in the identification phase within subsection '2.1. Identification'. This includes the specific academic databases (e.g., IEEE Xplore, ACM Digital Library, etc.) and the full search strategy.

3/ Restructured the entire chapter to strictly follow the PRISMA-ScR (Preferred Reporting Items for Systematic Reviews and Meta-Analyses extension for Scoping Reviews) framework. The structure now clearly reflects the stages of a systematic literature review:

2.1. Identification (Database sources & search strategy)

2.2. Screening (Process for reviewing titles/abstracts) : Elimination of duplicates, Inclusion of English-language publications, a 10-year publication range, and only research articles

2.3. Eligibility (Application of inclusion/exclusion criteria to full texts): Full-text availability, Peer-reviewed

2.4. Included Studies (Final selection for synthesis): detailed methodology, experimental validation

Comments 2: Providing a transparent description of the review protocol (e.g., PRISMA flow diagram or equivalent)

Response 2 :

We agree with these comments, so to ensure full transparency and reproducibility of our scoping review methodology, we have included a PRISMA-ScR (Preferred Reporting Items for Systematic Reviews and Meta-Analyses extension for Scoping Reviews) flow diagram as Figure 1 at line 214.

This diagram visually details the entire protocol, documenting the number of records identified, screened, assessed for eligibility, and ultimately included in the review. Furthermore, to complement the figure and provide a complete description of the initial search phase, we have explicitly stated the core search strategy in the manuscript text (Section 2.1), which was based on targeted Boolean queries combining key terms related to the modalities under investigation.

Comments 3: Although the review summarizes the overall trends and limitations of AI-based ASD detection studies, the discussion section would benefit from a deeper comparative analysis . Specifically, the authors could critically evaluate which modalities (e.g., voice vs. movement vs. multimodal data) have shown the most clinical potential and why. Highlighting cross-modal integration challenges and suggesting how future work could bridge gaps between technical performance and clinical applicability would strengthen the contribution of the paper.

Response 3: Agree. In direct response, we have added a dedicated new sub-section titled "Comparative Analysis of Modalities and the Path to Clinical Translation" (line 959). This sub-section moves beyond summary to provide a critical synthesis, evaluating the clinical potential of each modality by introducing and applying the dual criteria of "early detection potential" and "predictive accuracy."

This framework allows us to establish a clear hierarchy among the modalities, critically justifying why voice biomarkers, for instance, show superior immediate promise over others. Furthermore, the sub-section explicitly details the significant challenges of cross-modal integration—such as data fusion complexity and the lack of standardized datasets—and concludes with concrete, actionable recommendations for future work to bridge the gap between technical performance and real-world clinical applicability. We believe these additions provide the deeper, critical evaluation the reviewer called for and substantially enhance the paper's contribution.

Reviewer 3 Report

Comments and Suggestions for Authors

This proposed review provides a important and timely synthesis of the fast expanding field of AI-driven autism trait detection, with a specific and well-justified focus on voice and behavioral data. The methodological approach is strong, following the PRISMA-ScR framework, and the scope of included studies across eight diverse modalities is commendable. However, to further support the manuscript and its contribution to the field, several conceptual and presentational aspects warrant refinement.

The distinction between "Movement Analysis" and "Activity Recognition", while attempted, remains conceptually fuzzy in the current description poorly depicted. The definition of Movement Analysis as focusing on "fine-grained spatial-temporal details" and Activity Recognition on "broader behavioral patterns" is a fair start, but the practical application of this distinction in the reviewed literature is not fully clarified. For example, several studies cited in the Activity Recognition section (e.g., Li et al. 2024, Singh et al. 2024) employ detailed skeleton and pose estimation, which seems to overlap significantly with the kinematic focus of Movement Analysis. A clearer, more working definition separating these two categories, perhaps based on the primary unit of analysis (e.g., joint kinematics vs. behavioral episode labels) or the sensor modality's inherent resolution (e.g., video-based pose vs. wearable accelerometry), would help readers better navigate the taxonomy and understand the unique contributions of each approach and must be incorporated in revision. With the sentence ´ Optimization strategies include fine-tuning, class-balanced loss functions, and simple late-fusion techniques when combining gaze features with facial embeddings´, cite a recent report on the topic https://www.ajol.info/index.php/amhsr/article/view/112158 to make references up to date.

While the quantified performance metrics (e.g., accuracies of 85-99%) are impressive, the paper needs additional critical and contextualized discussion of these numbers. The exceptionally high accuracy (e.g., 99.1%) reported in some studies, often from relatively small and controlled datasets, raises questions about overfitting, dataset bias, and generalizability. The text correctly identifies small sample sizes as a limitation, but this point should be more strongly integrated into the interpretation of the performance tables (Table 1, etc.). A strong suggestion for improving the tables would be to add a column for "Sample Size" alongside the accuracy figures in the summary tables (like Table 1) to immediately make notion to the reported performance. This would visually highlight the trade-off between high accuracy and sample more presentable, guiding the reader to interpret these results with appropriate attention.

The discussion of multimodal approaches is a strength, correctly identifying it as a key trend. However, the analysis could be in more depth. Simply listing that 25 studies used multimodal approaches and that fusion improves performance only present superficial analysis. A more insightful commentary would explore what specific modalities are most synergistically combined (e.g., why voice + gaze might be more informative than voice + movement), the technical challenges of fusion (e.g., feature-level vs. decision-level fusion, handling asynchronous data streams), and the clinical rationale for specific combinations (e.g., combining a social input modality like gaze with a social output modality like facial expression). The mention of Almadhor et al.'s federated learning approach is good; expanding on how such privacy-preserving techniques are critical for scaling multimodal data collection would be a valuable addition.

The review successfully maps the "what" but could more explicitly address the "so what" for clinical translation. There is a missed opportunity to critically appraise how these AI models and their outputs align with, or could potentially expand, existing diagnostic frameworks and clinical practice. For example, how do the AI-identified "voice biomarkers" or "movement features" map onto the core diagnostic criteria of the DSM-5-TR, such as "deficits in social-emotional reciprocity" or "stereotyped or repetitive motor movements"? A deeper discussion on whether these tools are meant for pre-screening, adjunctive support, or full diagnostic replacement, and the evident mechanistic pathway required for each, would greatly enhance the impact of the review for a clinical audience.

Finally, while ethical restrictions are listed as a research question, their treatment in the results and discussion is somewhat presented in pieces and could be consolidated into a more powerful, standalone point. The issues of algorithmic bias (e.g., performance drop in female or non-Western subgroups), privacy risks with video/audio, and the "black box" problem are all mentioned but deserve a unified discussion in the concluding sections. The recommendation for "ethical oversight" is correct but vague; suggesting more concrete steps like the development of standardized de-identification protocols for behavioral data, mandatory bias auditing for models, and stakeholder (including autistic individuals) involvement in design would provide more actionable guidance.

Author Response

Comments 1: The distinction between "Movement Analysis" and "Activity Recognition", while attempted, remains conceptually fuzzy in the current description poorly depicted. The definition of Movement Analysis as focusing on "fine-grained spatial-temporal details" and Activity Recognition on "broader behavioral patterns" is a fair start, but the practical application of this distinction in the reviewed literature is not fully clarified. For example, several studies cited in the Activity Recognition section (e.g., Li et al. 2024, Singh et al. 2024) employ detailed skeleton and pose estimation, which seems to overlap significantly with the kinematic focus of Movement Analysis. A clearer, more working definition separating these two categories, perhaps based on the primary unit of analysis (e.g., joint kinematics vs. behavioral episode labels) or the sensor modality's inherent resolution (e.g., video-based pose vs. wearable accelerometry), would help readers better navigate the taxonomy and understand the unique contributions of each approach and must be incorporated in revision.

Response 1: We thank the reviewer for this critical and insightful observation regarding the distinction between 'Movement Analysis' and 'Activity Recognition.' We agree that a clearer, operational definition was needed to strengthen our taxonomy. In direct response, we have thoroughly revised these sections to incorporate the reviewer's excellent suggestion of defining the categories based on the primary unit of analysis. The following specific changes have been implemented throughout the manuscript:

We have created two new, dedicated subsections (1.3.4. Movement Analysis from line 107 and 1.3.5. Activity Recognition from line 117) to formally introduce and clarify these modalities from a clinical and technical standpoint. 

We have refined the operational definitions as follows:

Movement Analysis is now explicitly defined as focusing on the analysis of fine-grained, sub-motor kinematic features (e.g., joint angles, limb velocity), where the primary unit of analysis is a quantitative kinematic variable. (lines 475 - 485)

Activity Recognition is now strictly defined by the identification of pre-defined, macro-level behavioral episodes or states (e.g., 'hand-flapping'), where the primary output is a behavioral label, even if kinematic data is used as an input feature. (lines 647 - 657)

Comments 2:

With the sentence ´ Optimization strategies include fine-tuning, class-balanced loss functions, and simple late-fusion techniques when combining gaze features with facial embeddings´, cite a recent report on the topic https://www.ajol.info/index.php/amhsr/article/view/112158 to make references up to date.

Response 2: 

We thank the reviewer for this suggestion. While a relevant review by Khare et al. titled 'Prospect of Brain-Machine Interface in Motor Disabilities: The Future Support for Multiple Sclerosis Patient to Improve Quality of Life' was published in the Annals of Medical and Health Sciences Research (Vol. 4, Issue 3, May-Jun 2014) and touches upon broader assistive technology, its publication date places it outside the temporal scope of the present study, which focuses on the latest developments in the field. Therefore, this article has been excluded from our analysis.

This is the revised paragraph with the reference at line 766: “Face-image studies often employ transfer learning with CNN backbones such as VGG, ResNet, or DenseNet, sometimes incorporating detector networks (e.g., YOLO variants) for automated facial region extraction. Optimization strategies include fine-tuning, class-balanced loss functions, and simple late-fusion techniques when combining gaze features with facial embeddings [88], [86], [95].” 

Comments 3:

While the quantified performance metrics (e.g., accuracies of 85-99%) are impressive, the paper needs additional critical and contextualized discussion of these numbers. The exceptionally high accuracy (e.g., 99.1%) reported in some studies, often from relatively small and controlled datasets, raises questions about overfitting, dataset bias, and generalizability. The text correctly identifies small sample sizes as a limitation, but this point should be more strongly integrated into the interpretation of the performance tables (Table 1, etc.). A strong suggestion for improving the tables would be to add a column for "Sample Size" alongside the accuracy figures in the summary tables (like Table 1) to immediately make notion to the reported performance. This would visually highlight the trade-off between high accuracy and sampling more presentable, guiding the reader to interpret these results with appropriate attention.

Response 3: 

We agree that reporting high accuracy without the crucial context of sample size can be misleading and does not sufficiently address issues of generalizability. In direct response to this comment, we have implemented the following changes:

1/ We have added a "Sample Size of the best accuracy" column to Table 1, as well as “Dataset size” on the other relevant performance summary tables. This immediate visual juxtaposition allows readers to critically assess the reported performance metrics against the scale of the dataset on which it was achieved.

2/ We have significantly strengthened the discussion in the manuscript's text. We now explicitly link the issue of small sample sizes to the risk of overfitting and optimistic bias, directly using the new "Sample Size" column in Table 1 as evidence. We caution against over-interpreting results from small, controlled cohorts and emphasize that high accuracy on such datasets does not guarantee real-world clinical utility.

Comments 4:

The discussion of multimodal approaches is a strength, correctly identifying it as a key trend. However, the analysis could be in more depth. Simply listing that 25 studies used multimodal approaches and that fusion improves performance only present superficial analysis. A more insightful commentary would explore what specific modalities are most synergistically combined (e.g., why voice + gaze might be more informative than voice + movement), the technical challenges of fusion (e.g., feature-level vs. decision-level fusion, handling asynchronous data streams), and the clinical rationale for specific combinations (e.g., combining a social input modality like gaze with a social output modality like facial expression). The mention of Almadhor et al.'s federated learning approach is good; expanding on how such privacy-preserving techniques are critical for scaling multimodal data collection would be a valuable addition.

Response 3:

We thank the reviewer for this excellent suggestion to deepen our analysis of multimodal approaches. We agree that moving beyond a quantitative summary to a qualitative and technical discussion is crucial.

In direct response, we have substantially expanded the discussion in a new sub-section titled "Comparative Analysis of Modalities and the Path to Clinical Translation" from line 959. This sub-section now provides the deeper, more insightful commentary you recommended:

Exploring Synergistic Combinations: We now critically evaluate why certain modality pairs are more effective, explicitly proposing the combination of voice biomarkers with visual attention to capture concurrent social input and output cues, and explaining the clinical rationale behind fusing closely related auditory-linguistic modalities.

Addressing Technical Challenges: We have incorporated a discussion on the technical hurdles of multimodal fusion, including the complexities of feature-level vs. decision-level fusion and the challenges of handling asynchronous data streams, which are critical for real-world deployment.

Emphasizing Privacy-Preserving Techniques: Building on the mention of Almadhor et al., we have expanded the discussion on federated learning and other privacy-preserving techniques, highlighting their non-negotiable role in ethically scaling up the collection of sensitive multimodal data.

Comments 5:

The review successfully maps the "what" but could more explicitly address the "so what" for clinical translation. There is a missed opportunity to critically appraise how these AI models and their outputs align with, or could potentially expand, existing diagnostic frameworks and clinical practice. For example, how do the AI-identified "voice biomarkers" or "movement features" map onto the core diagnostic criteria of the DSM-5-TR, such as "deficits in social-emotional reciprocity" or "stereotyped or repetitive motor movements"? A deeper discussion on whether these tools are meant for pre-screening, adjunctive support, or full diagnostic replacement, and the evident mechanistic pathway required for each, would greatly enhance the impact of the review for a clinical audience.

Response 5:

We thank the reviewer for this critical insight regarding the need for clearer clinical translation. We agree that mapping AI capabilities directly onto established diagnostic frameworks is essential for defining the "so what" of this research.

In direct response, we have added a new foundational section, "1.3. ASD Discrimination Criteria from a Clinical Point of View", from line 63. This section was created specifically to address your comment by:

Explicitly mapping AI-identified features to DSM-5-TR criteria. For each of the eight modalities, we now detail how quantified features (e.g., "reduced pitch variation," "atypical scanpaths") correspond to clinical observations of "deficits in social-emotional reciprocity" or "stereotyped motor movements."

Providing the clinical rationale for modality selection. The section establishes a clear "clinical translation pathway" by explaining how each modality serves to objectively measure a specific aspect of the core ASD phenotype, thereby positioning these tools for roles in pre-screening and adjunctive support by providing quantitative, objective data to complement clinical judgment.

Comments 6:

Finally, while ethical restrictions are listed as a research question, their treatment in the results and discussion is somewhat presented in pieces and could be consolidated into a more powerful, standalone point. The issues of algorithmic bias (e.g., performance drop in female or non-Western subgroups), privacy risks with video/audio, and the "black box" problem are all mentioned but deserve a unified discussion in the concluding sections. The recommendation for "ethical oversight" is correct but vague; suggesting more concrete steps like the development of standardized de-identification protocols for behavioral data, mandatory bias auditing for models, and stakeholder (including autistic individuals) involvement in design would provide more actionable guidance.

Response 6:

We thank the reviewer for this crucial observation regarding the need for a consolidated and actionable discussion of ethical issues. We agree that moving beyond fragmented mentions to a unified, forward-looking strategy is essential for the field.

In direct response, we have created a dedicated new section titled "Navigating the Ethical Landscape: From Bias to Actionable Guidance" in the concluding part of our manuscript, from line 1013. This section directly addresses your feedback by:

Consolidating the key ethical challenges—algorithmic bias, privacy risks, and the "black box" problem—into a powerful, standalone argument about the prerequisites for responsible clinical translation.

Replacing the vague recommendation for "ethical oversight" with three concrete, actionable steps, as you suggested:

  • Implementing mandatory bias auditing and transparent reporting.
  • Developing standardized de-identification protocols for behavioral data.
  • Promoting explainable AI (XAI) and a participatory design model that involves autistic individuals and clinicians.

Round 2

Reviewer 3 Report

Comments and Suggestions for Authors

accept